# Advances on Natural Abietane, Labdane and Clerodane Diterpenes as Anti-Cancer Agents: Sources and Mechanisms of Action

**DOI:** 10.3390/molecules27154791

**Published:** 2022-07-26

**Authors:** Rosaria Acquaviva, Giuseppe A. Malfa, Monica R. Loizzo, Jianbo Xiao, Simone Bianchi, Rosa Tundis

**Affiliations:** 1Department of Drug and Health Sciences, University of Catania, Viale A. Doria, 95125 Catania, Italy; racquavi@unict.it (R.A.); bianchi.simone96@gmail.com (S.B.); 2CERNUT, Research Centre on Nutraceuticals and Health Products, Department of Drug and Health Sciences, University of Catania, Viale A. Doria, 95125 Catania, Italy; 3Department of Pharmacy, Health and Nutritional Sciences, University of Calabria, 87036 Rende, Italy; monica_rosa.loizzo@unical.it (M.R.L.); rosa.tundis@unical.it (R.T.); 4Nutrition and Bromatology Group, Department of Analytical and Food Chemistry, Faculty of Sciences, Universidade de Vigo, 32004 Ourense, Spain; jianboxiao@yahoo.com

**Keywords:** plant secondary metabolites, phytochemicals, terpenoids, tumor, cytotoxicity, apoptosis

## Abstract

Extensive research over the past decades has identified numerous phytochemicals that could represent an important source of anti-cancer compounds. There is an immediate need for less toxic and more effective preventive and therapeutic strategies for the treatment of cancer. Natural compounds are considered suitable candidates for the development of new anti-cancer drugs due to their pleiotropic actions on target events with multiple manners. This comprehensive review highlighted the most relevant findings achieved in the screening of phytochemicals for anticancer drug development, particularly focused on a promising class of phytochemicals such as diterpenes with abietane, clerodane, and labdane skeleton. The chemical structure of these compounds, their main natural sources, and mechanisms of action were critically discussed.

## 1. Introduction

Cancer is one of the principal causes of mortality and morbidity around the globe and the number of cases is constantly increasing and estimated to be 21 million by 2030 [1]. Currently, nearly one in six deaths worldwide is due to cancer [2] and, according to the World Health Organization (WHO), it is the second leading cause of death in the world, with 8.8 million deaths in 2015, 70% of which are in low-income countries or intermediate and, with an increase in the number of new cancer cases, about 70% over the next twenty years. In cancer, the uncontrolled normal cell proliferation produces genetic instability and alterations accumulate inside cells and tissues, transforming normal cells into malignant cells. This genetic instability includes mutations in tumour suppressor genes, DNA repair genes, oncogenes, and genes that are involved in cell growth. Causes of cancer are both intrinsic and extrinsic. The first group of causes is represented by biological factors such as genetic mutations and immunologic and endocrine disorders, whereas the second is represented by physical, chemical, or biological external factors such as radiations, smoking, certain metals, and infectious agents, including chemicals and pollutants.

The most common cancer types among men are prostate, colon, lung, rectum, and urinary bladder cancers. Among women the most five common types are the cancers of the breast, colon, rectum, uterine, thyroid, and lung [3].

Existing anticancer drugs affect also normal cells during the treatments, with serious side effects including cardiotoxicity, neurotoxicity, and immunosuppression, etc. Furthermore, the resistance of cancer cells to available drugs makes it necessary to identify new molecules with antitumour activity and less toxic effects. Secondary metabolites from the plant kingdom are promising sources of molecules with the potential of improving anti-cancer therapies.

For thousands of years, natural products have played an important role in preventing and treating several human diseases. Natural medicines have come from various source including terrestrial plants, marine organisms, and terrestrial microorganisms. In particular, plants have played a central role in the development of some traditional medicine systems. The importance of plants in the healthcare of many different cultures has been widely documented as well as their role in the treatment of cancer.

More than 3000 plant species have been described as remedies for the treatment of cancer and numerous plant extracts and isolated compounds are tested on various human cancer cell lines and experimental animals after purification and then sent to clinical trials [4,5,6,7,8,9,10,11]. In fact, over the past years, the great evolution of medical research has led to important advances in understanding of the genetic and molecular alterations that underlie cancer development and progression.

This has stimulated considerable research activities also in the area of natural products. Moreover, with improved molecular and cellular experimental systems, remarkable progress has been made in unravelling the mechanisms that underlie the anti-cancer properties of pure biomolecules.

The purpose of this review article was to highlight the potential anti-cancer activities and sources of a natural class of diterpenes with abietane, labdane, and clerodane skeletons belonging to the group of furanoditerpenoids. This particular group of rare diterpenoids presents a furan ring, which confers a significant biological activity due to the aromatic system that allows the formation of hydrogen bonds and hydrophobic interactions with cellular components.

## 2. Study Design

The available information was collected from some scientific databases such as SciFinder, PubMed, Science Direct, Scopus, and Web of Science, using the keywords “cancer” or “lung cancer” or “prostate cancer” or “breast cancer” or “pancreatic cancer” or “gastric cancer” or “cervical cancer” or “melanoma” and “diterpenes” or “labdane” or “clerodane” or “abietane” or “Lamiaceae” or “carnosic acid” or “tanshinone I” or “tanshinone IIA” or “sclareol” or “andrographolide” or “royleanone” or “columbin” or “clerodin”. The interest in diterpenes as potential anti-cancer agents is demonstrated by the presence of numerous studies in the literature. In a previous review, Islam [12] reported articles that investigated diterpenes and their derivatives as potential anti-cancer agents from January 2012 to January 2017. Herein, datasets were outlined from January 2017 to March 2022. However, some interesting works published before 2017 were also reported in order to critically comment on the reported data and evaluate the most active compounds and the possible structure-activity relationships. To confine the search, only English language articles were included in this study. To find relevant studies, papers were primarily screened based on titles and abstracts. 

Inclusion criteria were studies on diterpenes isolated from plants, in vitro, ex vivo, and in vivo studies, studies with diterpenes or their preparations and/or diterpenes associations with anticancer drugs, and studies with or without proposing mechanisms of action. Exclusion criteria were articles evaluating the anticancer activity of total extracts or fractions instead of purified constituents, studies on synthetic diterpenes, or studies non-covering the current topic.

## 3. Generalities of the Diterpenes

Diterpenes are a class of C20 natural compounds widely distributed in nature, which originate by condensation of four isoprene units. Depending on their skeletal core, this group of secondary metabolites can be classified in linear, bicyclic, tricyclic, tetracyclic, pentacyclic, and macrocyclic compounds. In recent years, diterpenes have attracted growing attention because of their interesting biological properties including anticancer activity. 

Various aspects of the great class of the diterpenes have been covered in different review articles [13,14,15,16,17]. Among them, González [14] analysed the biological properties of natural abietane-type diterpenes characterised by an aromatic C ring and reported the synthetic studies of this group of diterpenes. Previously, Topçu and Gören [16] described not only the cytotoxic and antitumour properties but also the cardiovascular, antimicrobial, insecticidal, and anti-leishmanial effects and other activities of diterpenes from Turkish Lamiaceae species.

## 4. Abietane Skeleton

Abietane diterpenes, naturally occurring compounds isolated from several plants, exhibited a wide variety of bioactivities, including anti-inflammatory, anti-microbial, anti-viral, and immunosuppressive activities [14]. However, these compounds deserve particular attention because of their anti-proliferative activity against several human cancer cell lines including breast (MDA-MB-231 and MCF7), liver (HepG2), lung (A549, NCI-H460, NCI-H460/R), pancreatic (MIA PaCa-2), colon (HCT116), endometrial carcinoma (HEC-1-A), and prostate (LNCaP) cell lines [18,19,20].

Fronza et al. [19] investigated five diterpenes isolated from *Peltodon longipes* (Lamiaceae) namely royleanone (**1**), 7*α*-acetoxyroyleanone (**2**), 7-ketoroyleanone (**3**), sugiol (**4**), and horminone (**5**) (Figure 1), against the human pancreatic cancer cells MIA PaCa-2.

Among them, 7*α*-acetoxyroyleanone (**2**) exhibited the highest activity with a half maximal inhibitory concentration (IC_50_) value of 4.7 μM. The other diterpenes showed a moderate cytotoxic activity with IC_50_ values in the range 32.5–17.9 μM. Taking into account that the cytotoxic properties of quinones are frequently elucidated by the ability of these compounds to covalently bind to proteins, DNA, and RNA, the alkylating properties of isolated diterpenes were examined by an extracellular system using 4-(4-nitrobenzyl) pyridine (NBP). As positive control, benzoquinone was used *p*- and showed the highest reactivity. 

Sugiol (**4**), also isolated from the bark of *Taxodium distichum* L. Rich var. distichum (bald cypress) by Zaher et al. [21] together with other diterpenes such as 6-α-hydroxysugiol, cryptojaponol, and 6-hydroxy-5,6-dehydrosugiol, showed a potent cytotoxic activity against human pancreas epithelioid carcinoma (PANC-1) cells adapted to nutrient-starved conditions (half-maximal effective concentration (EC_50_) value of 9.0 μM).

Royleanone (**1**), 7*α*-acetoxyroyleanone (**2**), and horminone (**5**) exhibited similar reactivity towards NBP. Consequently, since these diterpenes have different cytotoxic activities, the authors concluded that their cytotoxicity can only partially be attributed to their alkylating properties and further mechanisms are certainly involved. For this reason, the inhibitory activity against both human topoisomerases I and II of compounds **1–5** was also investigated. The most active compounds as inhibitors of the human topoisomerase I were 7-ketoroyleanone (**3**) and sugiol (**4)** (IC_50_ values of 2.8 and 4.7 μM, respectively), which showed greater inhibitory activity than the positive control camptothecin (IC_50_ of 28.0 μM). Both compounds also showed an interesting inhibitory activity against topoisomerase II, but with higher IC_50_ of 26.7 and 26.0 μM, respectively. 

Royleanone (**1**) is one of the most investigated abietane-type diterpenes. Recently, the anti-cancer activity of **1** was assessed against prostate cancer cell line (LNCaP) by Cell Counting Kit-8 (CCK8) test [22]. An IC_50_ value of 12.5 μM was found. Royleanone **(1)** also demonstrated induction of apoptosis in a concentration-dependent manner and induced G2/M cell cycle arrest in LNCaP cells in a concentration-dependent manner. mTOR/PI3/AKT is an important signalling pathway implicated in tumourigenesis and progression of different cancers. Royleanone (**1**) inhibited the expression of some of the important proteins involved in this pathway including p-AKT, p-PI3K, and p-mTOR.

In a successive study, some other royleanone-type diterpenes, such as 7*α*, 6*β*-dihydroxyroyleanone (**6**), 7*α*-formyloxy-6*β*-hydroxyroyleanone (**7**), 7*α*-acetoxy-6*β*-hydroxyroyleanone (**8**), and coleon U (**9**) (Figure 1) isolated from *Plectranthus madagascariensis* (Lamiaceae) were investigated against several human cancer cell lines such as colon (HCT116), breast (MDA-MB-231, MCF-7), and lung (NCI-H460, NCI-H460/R) cancer [23]. All compounds showed a growth inhibitory activity in most of the cells tested. Compounds **6** and **8** showed similar growth inhibition of the NCIH460 cancer cells (concentration causing 50% cell growth inhibition (GI_50_) of 25 and 2.7 μM, respectively) and its multidrug-resistant variant NCIH460/R (GI_50_ of 25 and 3.1 μM, respectively), suggesting that both ditepenes are not substrates for such efflux pumps.

Coleon U (**9**) was the most active against MCF-7 cells with a GI_50_ of 5.5 μM, followed by 7α-acetoxy-6β-hydroxyroyleanone (GI_50_ of 6.4 μM). MDA-MB-231 cells growth was not particularly affected by the tested compounds. A marked activation of caspases-3 and -9 by 6,7-dehydroroyleanone (**10**) (Figure 1) was reported by Garcia et al. [24]. The analysis of structure–activity relationships (SARs) for royleanone-type diterpenes revealed that the presence of an electron-donating group at C6 and/or C7 positions in the abietane skeleton seems to be crucial for cytotoxic effects. The cytotoxic properties of these compounds against several different cancer cells and the selectivity of the royleanone-type diterpenes mainly for non-small lung cancer cell lines may encourage further studies to prospect future applications as chemopreventive and/or chemotherapeutic agents. Successively, Sitarek et al. [25] isolated 6,7-dehydroroyleanone (**10**) from *Plectranthus madagascariensis* and *P. ecklonii* (Lamiaceae) together with other abietane diterpenes and investigated its potential activity in initiating apoptosis in a glioma cell line. 

6,7-Dehydroroyleanone (**10**) demonstrated to be able to induce apoptosis G2/M cell cycle arrest and double-strand breaks. Moreover, it was able to influence the expression of pro-apoptotic and anti-apoptotic genes, such as Bcl-2, Bax, Cas-3, or TP53. 

Tanshinones are a group of lipophilic abietane-type norditerpenoids compounds that represent the major bioactive constituents of the rhizome of the Chinese medicinal herb, *Salvia miltiorrhiza* (Lamiaceae). The chemical structure of tanshinone I (**11**), tanshinone IIA (**12**), and cryptotanshinone (**13**) is reported in Figure 2.

Tanshinone I (**11**) is an interesting diterpene that has been reported to exert anti-proliferative effects against different types of cancers. Li et al. [26] studied the mechanism of action of tanshinone I (**11**) on human endometrial cancer HEC-1-A cells, revealing that the diterpene was able to inhibit the proliferation of HEC-1-A cells (IC_50_ value of 20 μM). Tanshinone I (**11**) exerted anti-proliferative effects through the induction of apoptosis. Moreover, it produced an increase of reactive oxygen species (ROS) levels that are related to the reduction of the mitochondrial membrane potential levels and modulated the expression of JAK/STAT signalling pathway proteins. 

The anti-cancer properties of tanshinone IIA (**12**) has been confirmed by numerous research studies [27,28,29]. The diterpene showed a wide range of anticancer effects by inhibiting cancer growth, regulating cell cycle, inducing apoptosis, regulating signalling pathways, and reversing the multidrug resistance in several human cancer cell lines. Tanshinone IIA (**12**) inhibited the growth and proliferation of various cancer cells, including lung, colon, liver, and breast cancer, and leukemia [29]. 

Inhibition of cancer cell growth and proliferation is considered one of the main approaches for the treatment of various cancers. Generally, cell cycle arrest, induction of apoptosis, and endoplasmic reticulum (ER) stress can inhibit cell proliferation. Tanshinone IIA (**12**) activated ER-mediated apoptosis in human pancreatic cancer BxPC-3 cell-derived xenograft tumours by increasing protein expression levels of caspase-3, caspase-12, protein kinase RNA-like ER kinase (PERK), and ATF6, and by down-streaming eukaryotic initiation factor 2a (eIF2a) and C/EBP homologous protein [30]. 

Tanshinone IIA (**12**) inhibited the breast cancer cell line MCF-7 growth by inhibiting the mammalian target of rapamycin (mTOR), phosphatidylinositol-3-kinase (PI3K), mitogen-activated protein kinase (MAPK) signalling pathway protein kinase B (Akt), and protein kinase C (PKC) [31]. Moreover, **12** inhibited the adenosine monophosphate-activated kinase (AMPK), Parkin pathway, and S-phase kinase-associated protein 2 (Skp2), leading to the mitochondria-mediated apoptosis of cancer cells [32]. 

Tanshinone IIA (**12**) showed induction of S phase cell cycle arrest and apoptosis in lung cancer cells PC9 by regulating the PI3K-Akt signalling pathway [33]. 

Signal transducer and activator of transcription 3 (STAT3) is a member of signal-responsive transcription factors, plays a key role in tumourigenesis, and whose activation stimulates the expression of forkhead box M1 (FOXM1), a member of the FOX family whose over-expression promoted cancer progression and metastasis [34]. Tanshinone IIA (**12**) proved to suppress gastric cancer cells growth by the down-regulation of STAT3 and FOXM1 expression [35].

Moreover, **12** induced apoptosis in HeLa cells through mitochondria-dependent pathway and in lung cancer H1299 cells by murine double minute 4- inhibitor of apoptosis 3 (MDM4-IAP3) signalling pathway [36,37].

Angiogenesis plays an important role in the progression of cancer, infiltration, invasion, and metastasis. Tanshinone IIA (**12**) revealed antiangiogenic effects in vitro and in vivo in some cell lines, namely breast cancer, osteosarcoma, and vascular endothelial cell lines [38,39,40]. Vascular endothelial growth factor (VEGF) is a significant factor in the production and release of angiogenesis in cancer tissues under hypoxia. In human colorectal cancer cells, tanshinone IIA (**12**) restrained β-catenin/VEGF-mediated angiogenesis by targeting transforming growth factor-β (TGF-β1) in normoxic and hypoxia-inducible factor 1α (HIF-1α) in hypoxic micro-environments [41]. 

Some studies revealed that **12** inhibits both invasion and migration of colon cancer cells [42,43]. Tanshinone IIA (**12**) inhibited astrocytoma migration by the down-regulation of cellular-myelocytomatosis viral oncogene (c-Myc), Bcl-2, and matrix metalloproteinase-9 (MMP-9) expression, and the up-regulation of transmembrane receptor notch homolog 1 (Notch-1) pathway [44]. Moreover, in the gastric cancer cell line SGC7901, tanshinone IIA (**12**) down-regulated the expression of matrix metalloproteinase 2 (MMP-2), MMP-9, and FOXM1 [45].

The latest studies focused on autophagy have made progress in the understanding of the anti-cancer mechanisms of tanshinone IIA (**12**). Autophagy is an important physiological process that involves multiple pathways, such as the adenosine monophosphate-activated kinase (AMPK), the phosphatidylinositol-3-kinase (PI3K)/Akt, and the mammalian target of rapamycin (mTOR) signalling pathways [46,47]. Tanshinone IIA (**12**) suppressed colorectal cancer cell growth, decreased mitochondrial membrane potential, and inhibited mitophagy through inactivation of the AMPK/Skp1/Parkin pathway [32]. Other studies evidenced that **12** induced autophagy in human osteosarcoma cells [48] and in oral squamous cell carcinoma inactivated the PI3K/Akt/mTOR pathway and activated the Beclin-1/Atg7/Atg12-Atg5 pathway [49].

Recently, Li et al. [50] explored the mechanism of action of tanshinone IIA (**12**) in breast cancer stemness. Compound **12** reduced breast cancer cells stemness down-regulating the expression of stemness markers, inhibiting the spheroid formation, reducing the cancer-initiating ability, and decreasing the CD24-/CD44+ sub-population. Moreover, tanshinone IIA (**12**) enhanced the sensitivity of breast cancer cells to the chemotherapy drug doxorubicin. 

According to all these studies, tanshinone IIA (**12**) may be considered a promising compound and deserves further study for cancer therapy. 

The anti-cancer activity of cryptotanshinone (**13**) has been reported in several cancer cells including colorectal, bladder, melanoma, lung, liver, oesophageal, breast, ovarian, leukemia, gastric, renal, pancreatic, osteosarcoma, cervical, cholangiocarcinoma, leukemia, and rhabdomyosarcoma cancer cell lines [51,52,53,54,55,56,57]. At concentrations in the range of 2.5–40 μM, cryptotanshinone (**13**) was demonstrated to arrest in the G_0_/G_1_ phase human DU 145 prostate cancer and Rh30 rhabdomyosarcoma cell growth. 

Moreover, **13** inhibited the signalling pathway of the mammalian target of rapamycin (mTOR), and inhibited the expression of cyclin D1 and the phosphorylation of retinoblastoma protein. 

Similarly, in another work in a leukemia K562/ADM cell line, the ability of cryptotanshinone (**13**) was demonstrated, at concentrations ranging from 5 to 20 μM, to induce cycle arrest and apoptosis, with down-regulation of Bcl-2 and cyclin D1 [55]. Previously, Shin et al. [58] demonstrated that cryptotanshinone (**13**) acts as a selective activator and signal transducer of transcription 3 (STAT3) inhibitor. STAT3 plays a key role in tumourigenesis by transcriptionally up regulating its down-stream targets such as surviving, VEGF, and cyclin D1 that are involved in proliferation, angiogenesis, and apoptosis [59]. 

The inhibition of STAT3 as well as hypoxia inducible factors (HIF-1α) and MAPK/NF-kB signalling pathways are also involved in the suppression of colorectal cancer [60]. More recently, Fu et al. [61] reported that cryptotanshinone (**13**) selectively inhibited the growth and proliferation of SW620 and HCT116 cells while having little effect on SW480 cells in both in vitro and in vivo experiments. This cytotoxic effect is the consequence of a dual trigger of apoptotic and autophagic cell death. Moreover, it was demonstrated that **13** alone and in combination with other antineoplastic drugs targeted oncogenic mutations in both HT29 and HT116 colorectal cancer cell lines. This result was confirmed in vivo using mouse xenografts [62]. 

Gastric cancer (GC) is the second leading cause of tumour-related death. Approximately 90% of new cases of this cancer are related to the *Helicobacter pylori* infection and cytotoxin-associated gene A (CagA). Chen et al. [63] demonstrated that **13** inhibited the growth of gastric cancer cells in dose- and time-dependent manners. In particular, cryptotanshinone significantly inhibited the CagA-induced abnormal proliferation, migration, and invasion. The presence of IgG against CagA in the serum of GC patients was detected. The down-regulation of STAT3 and consequently the cell cycle arrest and stimulation of apoptosis was observed in pancreatic cancer cells [64]. Inhibition of Interleukin 6 (IL-6) STAT3 and consequently the induction of apoptosis was also observed in oesophageal cancer cells by Ji et al. [65]. The down-regulation of STAT3 is also involved in breast cancer inhibition induced by cryptotanshinone (**13**) [66]. 

More recently, Ni et al. [67] demonstrated that **13** inhibited the oligomer formation of Breast cancer resistance protein (BCRP) on the cell membrane, thus blocking its efflux function. This activity is strictly dependent on the expression level of estrogen receptor a (ERa) in ERa-positive breast cancer cells. However, **13** can also affect the proliferation of ERa-negative breast cancer cells characterised by a high expression of BCRP. Moreover, the combination of cryptotanshinone (**13**) and chemotherapeutic agents displayed enhanced antitumour activity. The breast cancer inhibitory effect is the consequence of the cell cycle arrest in G_2_/M phase and down-regulation of the expression of cyclin A, B, and D. A significant suppression of PI3K/AKT signal transduction was also observed [68]. 

The administration of cryptotanshinone (**13**) inhibits proliferation and growth of ovarian cancer cells by inhibition of glycolysis and down-regulation of glycolysis-related proteins including GLUT1, LDHA, and HK2 [69]. Moreover, **13** is able to inhibit the STAT3 signalling pathway to upregulate SIRT3. 

Several studies have demonstrated that prostaglandin E2 (PGE2) exerts a critical role on cancer cell progression. Cryptotanshinone (**13**) supplementation inhibits PGE2 and interferes with the PI3K/Akt signalling pathway and consequently interrupts hepatocellular carcinoma (HA22T) cell proliferation and cancer invasion [70]. To date, only one study investigated the effects of **13** on bladder cancer, one of the deadliest urothelial cancers. Liu et al. [71] demonstrated that cryptotanshinone (**13**) inhibits bladder cancer cell proliferation, migration, and invasion and induces apoptosis by a mechanism of action which involves the upregulation of the expression of PTEN that is responsible for the inhibition of PI3K/Akt/mTOR signal pathway in both 5637 and T24. 

Moreover, in melanoma cells, it elevates the expression of Bcl-2-like protein 4 (Bax) and at the same time decreases the Bcl-2 expression. The enhancing of ROS production as well as mitochondrial dysfunction was observed [72]. This abietane diterpene (**13**) was able to enhance the expression of onco-suppressor miR-133a with a decrease in proliferation and metastasis in lung cancer cells [73]. Moreover, **13** is an inhibitor of DNA topoisomerase 2 that regulates chromosomal segregation and DNA replication that has no effect on normal tissue [74].

Cryptotanshinone (**13**) suppresses onco-proliferative and drug-resistant pathways of chronic myeloid leukemia by targeting STAT5 and STAT3 phosphorylation in K562 cells [44].

It induces apoptosis by induction of caspase 3 and 7 and PARP and reduction of mitochondrial membrane potential. Patients with chronic myeloid leukemia frequently develop resistance to the Bcr-Abl tyrosine kinase inhibitors (TKIs) that limit the clinical applications of several drugs including imatinib. Cryptotanshinone (**13**) enhanced the antiproliferative activity of TKIs by inducing the cleavages of caspase and inhibiting STAT3 and eIF4E pathways activities. The administration of combination of **13** and imatinib induced a 2.6-fold higher suppression of tumour growth of xenografts than those animals treated with imatinib alone. This antiproliferative activity is the consequence of an increase of apoptosis [75]. 

Several cancer cells express, on their membranes, the death-receptor Fas. However, some cancer cells demonstrated to be resistant to Fas-mediated apoptosis since Fas activation can concurrently up-regulate Bcl-2 through p38 MAPK kinase and JNK pathways. Cryptotanshinone (**13**) (at concentrations of 1–10 μM) showed to sensitise DU145 prostate cancer cells to Fas-mediated apoptosis by the inhibition of the expression of Fas-mediated Bcl-2 [76]. DU145 cells are STAT3 highly active cells. Cryptotanshinone (**13**) (at the concentration of 7 μM) inhibited DU145 cells growth by suppression of STAT3 dimerisation, nuclear translocation, and DNA binding [58]. At the same concentration, tanshinone IIA (**12**) did not show inhibition of STAT3 activity, although another study reported that diterpene **12** at concentrations ranging from 3.4 to 27 μM inhibited STAT3 in rat C6 glioma cells [77]. This discrepancy may be ascribed to the different cell types investigated. However, it must take into account that compound **13** is converted into compound **12** in vivo; therefore, the anti-cancer activity in in vivo studies is likely the combined effects of both tanshinones. 

Carnosic acid (**14**) (Figure 2) is an abietane diterpene common in the Lamiaceae family. Since its first isolation from *Salvia officinalis*, several works have been published on plants, mainly *Salvia* and *Rosmarinus*, rich in carnosic acid (**14**). This diterpene was extensively investigated for its bioactivity, including its potential anti-cancer properties. Recently, Shao et al. [78] investigated the anti-cancer effects of carnosic acid (**14**) and its combination with temozolomide, an alkylating agent used as a treatment of some brain cancers, in glioma cancer cells. Carnosic acid (**14**) was shown to increase the cytotoxic activity of the alkylating agent, to increase cell cycle arrest and cellular apoptosis induced by temozolomide via activation of PARP and Caspase-3 and inhibition of cyclin B1, and to enhance the colony formation and cell migration inhibition induced by temozolomide. Moreover, carnosic acid (**14**) stimulated autophagy by p62 down-regulation, p-AKT inhibition, and LC3-I to LC3-II transition, induced by temozolomide.

Moreover, carnosic acid (**14**) has been the subject of numerous studies aimed at evaluating its possible association with other natural compounds and some known anticancer agents. Herein, we reported some of the recent works. Fisetin is a naturally occurring flavonoid known for interesting biological properties, including anticancer activity. Using in vitro and in vivo models, the potential benefits of the combination of carnosic acid (**14**) and fisetin on lung cancer were explored [79]. The combination of these compounds led to apoptosis in lung cancer cells with a promotion of caspase-3 signalling pathway, an increase of Bax and Bad pro-apoptotic signals, and a decrease of anti-apoptotic proteins of Bcl-2 and Bcl-xl. The death receptor of tumour necrosis factor-related apoptosis-inducing ligand (TRAIL) was enhanced in the treatment with the combination of carnosic acid (**13**) and fisetin. Furthermore, as demonstrated by mouse xenograft model, the combined treatment of **13** and fisetin was more active in inhibiting lung cancer cells growth than the monotherapy with **13** or fisetin. 

ERBB2 is a member of the epidermal growth factor (EGF) receptor family that is over-expressed in up to 20–30% of human breast cancer. D’Alesio et al. [80] explored the potential association of trastuzumab, a monoclonal antibody used for breast cancer therapy, and carnosic acid (**14**) on cell survival of ERBB2 over-expressing (ERBB2^+^) cells and whether compound **14** is able to restore trastuzumab sensitivity in cancer cells that have become resistant to the drug. The development of trastuzumab resistance remains an important challenge. 

In ERBB2^+^ cancer cells, compound **14** reversibly enhanced trastuzumab inhibition of cells survival, inhibited cell migration, and induced cell cycle arrest in the G_0_/G_1_ phase. PI3K/AKT/mTOR signalling pathway deregulation, ERBB2 down-regulation, and up-regulation of CDKN1B/p27KIP1 and CDKN1A/p21WAF1 were evidenced. 

NAD(P)H quinone oxidoreductase 1 (NQO1) is one of the two major quinone reductases in mammalian systems that plays multiple and important roles in cellular adaptation to stress. 

NQO1-dependent anti-cancer drugs such as β-lapachone are attractive candidates for cancer therapy because several cancers, including cervix, breast, pancreas, colon, and lung, exhibit higher expression of NQO1 than adjacent tissues. Arakawa et al. [81] reported the ability of carnosic acid (**14**) to induce, in melanoma cells, the expression of NQO1 through NF-E2 related factor 2 (NRF2) stabilisation, thus considerably enhancing the β-lapachone cytotoxicity. Carnosic acid (**14**) exhibited significant growth inhibition and cell cycle arrest at the G0/G1 phase and enhanced p21 expression in melanoma B16F10 xenograft cells [82]. Furthermore, **14** reduced, in vivo, the values of alanine aminotransferase (ALT) and aspartate aminotransferase (AST).

Carnosic acid (**14**) has also been suggested to be a potential anti-cancer agent in the treatment of cervical cancer, the fourth-most common cancer, killing many women in the world. Su et al. [83] demonstrated in vitro that compound **14** produced promotion of apoptosis progression via stimulating Caspase 3 expression, and ROS production with phosphorylation of (c-Jun *N*-terminal kinase (JNK) and its related signals. The development and growth of xenograft cancer in nude mice were inhibited by carnosic acid (**14**). Recently, El-Huneidi et al. [84] demonstrated that compound **14** inhibited human gastric cancer cells (AGS and MKN-45) proliferation in a dose-dependent manner. This cytotoxic activity is related to the inhibition of the phosphorylation/activation of Akt and mTOR. Moreover, carnosic acid (**14**) influenced apoptosis pathway by inducing the cleavage of PARP and the expression of caspases 3, 8, and 9. The cell cycle arrest at the G2/M phase as well as the promotion of apoptosis by inducing DNA damage, the inhibition of cell migration, and a coordinated inhibition of extracellular signal-regulated kinase (ERK), p-38, and JNK signalling pathways was observed in esophageal squamous cell carcinoma (KYSE-150), an aggressive type of esophageal cancer [85].

Carnosic acid (**14**) was also able to induce apoptosis in liver cancer cells [86]. However, its poor solubility and absorption severely limit its in vivo antitumour activity. For this reason, **14** was loaded into liposomes and further conjugated with transferrin (Tf-LP-CA) improve its bioactivity. In both HepG2 and SMMC-7721 cells, Tf-LP-CA was absorbed by liver cancer cells better than the acid alone and was able to promote apoptosis by inducing reduction of the mitochondrial membrane potential and enhancing the expressions of cleaved poly (ADPribose) polymerase, and caspase-3 and -9. This activity was confirmed in HepG2- and SMMC-7721-xenotransplanted mice. 

It can synergistically cooperate with curcumin to induce apoptosis in acute myeloid leukemia cells, but not in normal hematopoietic and non-hematopoietic cells [87,88]. A similar situation was also observed against DU145 and PC-3 metastatic prostate cancer cells where instead of cytotoxic activity, the combination curcumin+carnosic acid induced a G_0_/G_1_ cell cycle arrest together with a dissipation of the mitochondrial membrane potential, without generating oxidative stress, and was associated with defective oxidative phosphorylation encompassing mitochondrial dysfunction [89]. 

Carnosic acid (**14**) can inhibit human non-small cell lung carcinoma cells (A-549) growth in a concentration-dependent manner. The cytotoxic activity was mediated via apoptosis by suppressing PI3K/AKT/m-TOR signalling pathway. Moreover, compound **14** inhibited cell migration and invasion [90]. This terpene inhibited the human non-small-cell lung (NCI-H460) cancer proliferation via cell cycle arrest at G_0_/G_1_ and G_2_/M phases in a concentration-dependent manner. Although carnosic acid exerted antitumour activity against NCI-H460 cells, it induced toxic effects in non-tumoural cells, and thus needs to be considered carefully prior to pharmacological use therapeutically [91].

In conclusion, the data described above suggest that cell death induced by abietane diterpenes may not follow a single mechanism, but several mechanisms of action. The reported studies on abietane diterpenes against most of the cancer cell lines tested have shown that the interest of these compounds remains relevant and encourage the continuation of studies to increase knowledge regarding the cellular and molecular mechanisms of these interesting natural compounds for the development of new chemo-preventive, chemo-adjuvant, or chemotherapeutic agents. A summary of the main anti-cancer activities of abietane is shown in Table 1.

## 5. Labdane Skeleton

The interest in studying labdane-type diterpenes was intensified in the past decades due to a wide range of bioactivities, including anti-inflammatory, anti-microbial, anti-mutagenic, and cytotoxic properties [92,93,94]. The family of labdane-type diterpenes constitutes a significant class of natural products, being composed of over 7000 compounds, which have been isolated from plants of several families, such as Asteraceae, Acanthaceae, Lamiaceae, Cistaceae, Cupressaceae, Pinaceae, Annonaceae, Caprifoliaceae, Zingiberaceae, Solanaceae, Apocynaceae, and Verbenaceae [95].

Many of these compounds have been explored as potential antitumour agents. One of the most investigated compounds of this class of active biomolecules is andrographolide (**15**) (Figure 3), which revealed strong anticancer properties in several studies. In particular, andrographolide (**15**) has been reported to be able to inhibit cancer migration and invasion in different human cancer cell lines and models, and induce apoptosis and cell cycle arrest [96,97,98]. 

Andrographolide (**15**) is the dominant constituent of *Andrographis paniculata* (Acanthaceae). It was isolated also from *Swertia pseudochinensis* (Gentianaceae). Some labdane diterpenes with andrographolide-like skeleton have also been isolated from plants of Asteraceae, Cupressaceae, Lamiaceae, and Zingiberaceae families [99,100,101,102]. Several studies reported the potential use of compound **15** in the treatment of colorectal cancer, which is the third-most common cancer worldwide. 

5-Fluorouracil (5-Fu) is the most important drug used to treat this cancer. Taking into account that the resistance to the treatment with 5-Fu is a growing concern in colorectal cancer practice, Su et al. [103] explored the effects against colorectal cancer HCT-116 cells of combined 5-Fu and andrographolide (**15**). 

In vitro andrographolide (**15**) acted synergistically, enhancing the anti-proliferative effects of 5-Fu due to increased apoptosis. Andrographolide (**15**) enhanced the anti-cancer activity of 5-Fu by down-regulating the levels of phosphorylated cellular-mesenchymal to epithelial transition factor. These results confirmed the synergistic action of compound **15** on colorectal cancer cells.

Successively, Khan et al. [104] demonstrated that andrographolide (**15**) was able to decrease cell viability of colon cancer HT-29 cells in a concentration- and time-dependent manner and to induce apoptosis. These effects appeared to be linked with interference with mitochondrial membrane potential and increased levels of intracellular ROS. Moreover, it was found that the diterpene produced cell cycle arrest in G2/M phase at lower concentrations and in the G0/G1 phase at higher concentrations. 

The potential synergism between andrographolide (**15**) and another anti-cancer drug, namely topotecan, an analogue of the natural alkaloid camptothecin, was analysed in acute myeloid U937 cells [105]. Topotecan acts as a topoisomerase I inhibitor and it is used in the treatment of several cancer types including leukemia. Both andrographolide (**15**) and topotecan exhibited anti-proliferative properties in a concentration-dependent manner when applied separately. The cells pre-treatment with andrographolide before applying topotecan produced a synergistic activity. Moreover, the cell-cycle arrest at S phase and the up-regulation of the expression of pro-apoptotic proteins produced by topotecan is made more prominent upon pre-treatment combination with **15**. 

More recently, Gao et al. [106] assessed the effects on p53 and Mouse double minute 2 homolog (Mdm-2) pathways of andrographolide (**15**) on gastric carcinoma cells proliferation. Andrographolide (**15**) inhibited the proliferation of human SGC7901 and AGS cell lines with IC_50_ values 38 and 44 μM, respectively. Moreover, **15** activated the expression of the p53 protein and gene, and down-regulated the levels of the negative regulator of p53, Mdm-2.

The promising activity of andrographolide (**15**) was also demonstrated against breast cancer [107]. Andrographolide (**15**) was able to suppress cell proliferation, migration, and invasion of MCF-7 cancer cells in vitro, and to inhibit cancer growth and metastasis in MMTV-PyMT mice.

Moreover, **15** inhibited the expression of miR-21-5p and further promoted programmed Cell Death 4 (PDCD4) via nuclear factor kappa-light-chain-enhancer of activated B cells (NF-κB) suppression. 

The labdane diterpene sclareol (**16**) (Figure 3) isolated from *Salvia sclarea* (Lamiaceae) has demonstrated induction of cell cycle arrest and apoptosis in leukemic and human breast cancer cells [108,109,110,111]. Additionally, liposome-encapsulated sclareol was able to inhibit the human colon cancer growth xenografted in immunodeficient mice [110,111]. Zhang et al. [112] explored the effects of sclareol (**16**) in cervical carcinoma HeLa cells. Sclareol (**16**) inhibited cell proliferation in a concentration- and time-dependent manner and demonstrated to induce apoptosis. Moreover, to identify the potential anti-cancer target of **16**, the expression of several cancer-associated proteins was investigated. Superoxide dismutase 1 (SOD1) and caveolin 1 (Cav1) were identified as potential targets of sclareol (**16**). 

The treatment with **16** produced a decrease of the protein level of SOD1 in a time-dependent manner. The decreased expression of SOD1 was negatively correlated with the increased expression of Cav1. The role of Cav1 in tumourigenesis is controversial. However, the down-regulation of Cav1 has been reported in the development and progression of different cancer types cutaneous including lung, colon, and breast cancer. Even if the exact role of Cav1 in HeLa cells still has to be elucidated, Zhang et al. [112] showed that the up-regulation of Cav1 correlated with decreased cell viability under the treatment with sclareol (**16**). This indicated that Cav1 might function as a cancer suppressor in cervical carcinoma cells. Sclareol (**16**) was not able to induce cAMP increase, unlike its related compound forskolin, supporting the notion that compound **16** acts with a different mechanism of action than that of forskolin. Sclareol (**16**) induced apoptosis in human cancer cell lines and suppression of cancer growth [113,114,115,116]. 

The main obstacle to the use of **16** is its poor bioavailability due to its lipophilicity and poor aqueous solubility. In order to favour its administration in physiological media, and then to increase its activity and to obtain a stable formulation, recently Cosco et al. [117] realised nanoparticles with polyester poly-lactic-co-glycolic acid (PLGA) and sclareol (**16**). The diterpene was efficiently retained by the polymeric structure and did not induce any physical destabilisation. Equivalent activity on human breast cancer and colon cells with respect to the free drug was found. 

The coating of the PLGA nanoparticles with hyaluronic acid (1.5 MDa) increased the antitumour efficacy of the encapsulated drug against human breast cancer cells expressing the hyaluronan receptors (MCF-7 and MDA-MB468), while a similar pharmacological effect was obtained on human colon carcinoma cells (CaCo-2). In the same year, Borges et al. [118] assessed the synergistic action of sclareol (**16**) and doxorubicin in vitro in the human breast cancer cell line MDA-MB-231 and both in vitro and in vivo in the murine breast cancer (4T1) cells in their free form and co-loaded in nanostructured lipid carriers (NLC). The combination sclareol (**16**)/doxorubicin (molar ratio 1:1.9) notably increased the anti-cancer activity of doxorubicin in breast cancer cells either in their free form or in co-loaded in NLC. 

The anti-cancer activity and the potential mechanisms of sclareol (**16**) in human small cell lung carcinoma cell line (SCLC) were recently studied in vitro and in vivo in a xenografted cancer mice model [119]. Sclareol (**16**) significantly reduced cell viability, induced G_1_ phase arrest, and triggered apoptosis in the H1688 cell line. The growth arrest induced by **16** was associated with DNA damage as indicated by activation of ATR and Chk1 and H2AX phosphorylation. Moreover, in vivo sclareol (**16**) inhibited cancer weight and volume. 

Three new labdane diterpenes, namely hedylongnoids A (**17**), B (**18**), and C (**19**) (Figure 4), together with three known ones such as yunnancoronarin A (**20**) and hedyforrestin B (**21**) and C (**22**) (Figure 4), were isolated in 2015 from the rhizomes of *Hedychium longipetalum* (Zingiberaceae) by Zhao et al. [120], and their potential cytotoxic activity against Hela and SGC-7901 cancer cell lines was also assessed. 

Against HeLa cells, the most active compound was yunnancoronarin A (**20**) with IC_50_ values of 6.21 and 6.58 μg/mL, against SGC-7901 and HeLa cell lines, respectively. Interesting results were also obtained with hedyforrestin C (**22**) and hedylongnoid C (**19**) against SGC-7901 cells (IC_50_ values of 7.29 and 8.74 μg/mL, respectively. The new diterpenes 17 and 18 showed no cytotoxic activity (IC_50_ values > 25 μg/mL). 

A recent work led to the isolation of a new labdane diterpene, (1R,4aS,5R,8aS)-1,4a-dimethyl-5-[(1E)-3-oxobut-1-en-1-yl]decahydronaphthalene-1-carboxylic acid (**23**) (Figure 4), from *Juniperus oblonga* (Cupressaceae) together with other known compounds [121]. Isolated compounds were assessed for their cytotoxic effects against, MCF-7, HeLa, and HepG2 human cancer cell lines. However, this compound showed a weak cytotoxicity with percentage of inhibition at 4 mg/mL of 18.39%, 26.36%, and 28.45% against HepG2, HeLa, and MCF-7 cells, respectively.

### Ent-Labdane Skeleton

Six new ent-labdane diterpenes, uasdlabdanes A-F (**24–29**) (Figure 5), were isolated from the dichloromethane fraction of the ethanol extract of the aerial parts of *Eupatorium obtusissmum* (Asteraceae) [122]. Isolated compounds were investigated for their anti-proliferative activity against six human cancer cell lines such as HBL-100 (breast), T-47D (breast), HeLa (cervix), A549 (lung), SW1573 (lung), and WiDr (colon) cells. GI_50_ values in the range from 19 to >100 μM were found. Uasdlabdane D was the most active with a GI_50_ value of 19 μM against HeLa cells.

Interesting results were also obtained against A549 and T-47D (GI_50_ values of 23 and 24 μM, respectively). However, the diterpenes were less active than the positive control. A comparative anti-proliferative activity was found for uasdlabdane E (**27**) and F (**28**) against A549 and HeLa cell lines. Uasdlabdanes A–C (**24–26**) showed GI_50_ > 100 μM in all tested cell lines. 

The analysis of structure–activity relationships suggests that a C-7 carbonyl group causes loss of activity (see uasdlabdanes A–C); a C-7 acetoxy group determines the best anti-proliferative effects (see uasdlabdane D vs. uasdlabdane E and F). Recently, Balbinot et al. [95] investigated the chemical profile and the anti-proliferative activity of the aerial parts of *Grazielia gaudichaudeana* (Asteraceae). Fifteen compounds were identified and tested. Among these constituents, there were three 18-nor-ent-labdane diterpenes (**30–32**) (Figure 5). Compound **30** exhibited cytotoxic activity against ovary cell line (OVCAR-03) with a GI_50_ value of 5.5 μM and glioma cell line U251 with a GI_50_ value of 25.63 μM and compound **32** was selective to ovary cells (OVCAR-03) with a GI_50_ of 28.5 μM. A summary of the main anti-cancer activities of labdane diterpenes is shown in Table 2.

## 6. Clerodane Skeleton

The clerodane diterpenes are a large family of bicyclic secondary metabolites ubiquitous not only in hundreds of plant species from a number of different families but also in marine organisms, bacteria, and fungi. These metabolites result from the clerodane carbon skeleton that in turn derived from the shifting of methyl and hydride in labdane carbon skeleton. 

Based on the 5:10 *trans* or *cis* ring fusion, we may identify two main groups of compounds: the *trans* series from columbin (**33**) (Figure 6), which represents the 25% of derivatives, and the *cis* series from clerodin (**34**) (Figure 6) for the remaining 75% of compounds (*neo*-clerodanes). Compound **33** has been isolated in several plants such as *Spenocentrum jollyanum* Pierre (Menispermaceae) and *Jateorhiza columba* Miers (Menispermaceae) while compound **34** was found in *Clerodendrum infortunatum* L. (Lamiaceae). Further classification may be made also considering the relative configurations at C-8 and C-9, defining four more groups: *trans*-*cis* (TC), *trans*-*trans* (TT), *cis*-*cis* (CC), and *cis*-*trans* (CT) [123].

Subsequent coexisting and independently developed biosynthetic pathways produce the vast amount of clerodane compounds in the plant kingdom. As confirmed by the recent literature, there are a large number of species producing yet unidentified clerodanes. This family of diterpenes is known to exhibit remarkable pharmacological properties, in particular as potential anticancer agents. In the light of these facts, the research of new natural or synthetic clerodanes continues to be the object of numerous works.

A new clerodane diterpene, namely corymbulosin X (**35**) (Figure 7), constituted by an isozuelanin skeleton, was isolated from *Anacolosa clarkii* (Olacaceae). The compound showed significant antiproliferative and cytotoxic activities against four paediatric cancer cell lines: Ewing sarcoma (A-673), rhabdomyosarcoma (SJCRH30), medulloblastoma (D283), and hepatoblastoma (Hep293TT). The total growth inhibition (TGI) value for each cell line has been calculated with the sulforhodamine B (SRB) assay using the chemotherapeutic vinorelbine as positive control. 

Corymbulosin X (**35**) was the most active of all the clerodane compounds isolated form *Anacolosa clarkii*, for each tested cell line, with TGI values of 0.7 μM (A-673 cells), 0.34 μM (SJCRH30 cells), 0.36 μM (D283 cells), and 0.22 (Hep293TT cells).

Regarding hepatoblastoma, it seems that the C-6 hydroxy group contained in the compound **35** was important for conferring activity against this tumour cell line [124].

Caseakurzin B (**36**) (Figure 7) was isolated from twigs and leaves of *Casearia kurzii* (Salicaceae). The compound has shown anti-proliferative activity against lung epithelial carcinoma (A549) cell line, with IC_50_ value of 4.4 μM. The compound was also able to arrest the cell cycle at G_2_/M phase and to induce apoptosis on A549 treated cells (apoptosis rate: 38.97% at 20 μM) [125].

Corymbulosin M (**37**) (Figure 7) from *Casearia kurzii* has shown cytotoxic activities also on A549 (IC_50_ of 5.5 μM), HeLa (IC_50_ of 4.1 μM), and HepG2 (IC_50_ of 9.3 μM) cells. The compound was found to be more active than etoposide, used as positive control (IC_50_ value, respectively, 16.5, 25.8, and 16.0 μM). The presence of an acetyloxy group of C-18 seems to contribute positively to the cytotoxic effects. Regarding the mechanism of action, the compound was able to induce apoptosis on HeLa cells in a dose-dependent manner. The induction of apoptotic death appears to be related to the compound’s ability to arrest the cell cycle at the G_0_/G_1_ stage [126].

Kurzipene D (**38**) (Figure 7) from *Casearia kurzii* (Salicaceae) showed cytotoxic activity against four human cancer cell lines: HepG2 (IC_50_ of 9.7 μM), A549 (IC_50_ of 10.9 μM), HeLa (IC_50_ of 12.4 μM), and K562 (IC_50_ of 7.2 μM). For three of the four cell lines (HepG2, HeLa, and K562), the activity of the compound was greater than that of etoposide (used as a positive control), while on the fourth (A549) it was practically comparable. In fact, the IC_50_ values for etoposide were, respectively of 16.0 μM (HepG2), 10.4 μM (A549), 36.1 μM (HeLa) and 17.9 (K562). Regarding the mechanism of cell death, compound 38 was able to induce apoptosis (93.20% of apoptotic cell after a 48 h treatment with a concentration 27 μM) and to arrest the cell cycle at S phase in HepG2 cells. Furthermore, kurzipene D (**38**) has also been tested in vivo on zebrafish embryos, where the compound was able to block K562 cells proliferation and migration in a dose-dependent manner [127].

Casearin D (**39**) (Figure 7), caseargrewiin F (**40**), and casearin X (**41**) (Figure 8) isolated from *Casearia sylvestris* (Salicaceae) exerted significant cytotoxic activity against caucasian promyelocytic leukemia (HL-60), MDA-MB/231, human breast carcinoma (Hs578-T and MX-1), caucasian prostate adenocarcinoma (PC-3), DU-145 and murine melanoma (B-16/F10) cell lines. 

The IC_50_ values for Casearin D (**39**) were 3.44 μM (HL-60), 4.23 μM (MDA-MB/231), 4.39 μM (Hs578-T), 6.50 μM (MX-1), 1.41 μM (PC-3), 8.53 μM (DU-145), and 6.52 μM (B-16/F10), respectively.

Casearin X (**41**) was more potent, with IC_50_ values of 0.28 μM (HL-60), 1.51 μM (MDA-MB/231), 1.14 μM (Hs578-T), 0.95 μM (MX-1), 0.86 μM (PC-3), 1.19 μM (DU-145), and 1.15 μM (B-16/F10). Caseargrewiin F (**40**) was the most cytotoxic of the molecules treated in the study, with IC_50_ values of 0.20 μM (HL-60), 0.14 μM (MDA-MB/231), 0.26 μM (Hs578-T), 0.36 μM (MX-1), 0.31 μM (PC-3), 0.76 μM (DU-145), and 0.16 μM (B-16/F10). 

The anti-proliferative activity of these compounds was confirmed by the 5-bromo-2′-deoxyuridine (BrdU) assay, a test to assess the amount of new DNA synthesised by the cells. The test was carried out on HL-60 cells and the data obtained, collected after 24 h of exposure to the substances and expressed as a percentage of BrdU-positive cells, were 35.0% for casearin D (**39**) (4 μM), 32.7% for caseargrewiin F (**40**) (1 μM), and 30.8% for casearin X (**41**) (1.5 μM). Comparing these results with those of the negative control (66.4%), it is clear that the substances caused a reduction in BrdU incorporation and, consequently, the synthesis of new DNA, confirming the anti-proliferative action. Further tests on HL-60 cells were carried out to assess the mechanisms responsible for the anti-proliferative action.

The analysis of cell membrane integrity showed that, after a 24 h treatment, casearin D (**39**) (4 μM) and caseargrewiin F (**40**) (1 μM) lowered the integrity value to 71.8% and 74.9%, while casearin X (**41**) (1.5 μM) was less effective, reaching only 94.1%. All compounds caused DNA fragmentation in a concentration-dependent and time-dependent manner. After a 24 h treatment, casearin X (**41**) (1.5 μM) led to 25.4% DNA fragmentation, caseargrewiin F (**40**) (1 μM) to 44.2%, and casearin D (**39**) (4 μM) to 33.2%. Furthermore, casearin X (**41**) (0.7 μM; 1.5 μM) was able to arrest the cell cycle at G_0_/G_1_ phase after a 24 h exposure and both casearin X (**41**) and caseargrewiin F (**40**) were able to activate caspase-8 and caspase-9, inducing apoptotic death in the treated cells. This was confirmed by observing the ability of these two compounds to cause phosphatidylserine externalisation and morphological changes typical of apoptotic death [128]. Another study demonstrated that casearin X (**41**) was an efficient in vivo antitumour substance by reducing the proliferation of neoplastic cells in adult female Swiss mice (*Mus musculus*) with Sarcoma 180 tumour. The doses 25 mg/kg/day (intraperitoneal) and 50 mg/kg/day (oral), administered for 7 consecutive days, inhibited the tumour growth, respectively, by 90.0% and 65.5% [129]. Casearin D (**39**) was also found to be cytotoxic on HepG2 (IC_50_ of 6.04 μg/mL). The compound arrested the cell cycle at the G_0_/G_1_ phase, and the authors of the study showed that this was due to its ability to reduce ERK phosphorylation and Cyclin D1 expression [130]. 

Another clerodane diterpene from *Casearia sylvestris* named casearin J (**42**) (Figure 8) has been shown to induce apoptosis in T-cell acute lymphoblastic leukemia (T-ALL) cells and this study thoroughly investigated the mechanisms underlying this effect. The compound was able to inhibit the sarcoendoplasmatic reticulum calcium ATPase (SERCA) pump and to induce the depletion of the endoplasmic reticulum calcium reserves. In human acute lymphoblastic leukemia T-lymphoblasts (CCRF-CEM) cells the calcium loss was greater, while the human acute lymphoblastic leukemia drug-resistant T-lymphoblasts (CEM-ADR5000) cells suffered less from the effect of the substance due to their higher resistance against the SERCA pump inhibitor. The release of calcium from its storage site causes several damaging effects on the cell, inducing oxidative stress, reduction in Notch1 expression on the cell surface, and cell death by disrupting intracellular Ca^2+^ homeostasis (in CCRF-CEM cells, IC_50_ of 0.7 μM). Compound **42** also showed the ability to activate procaspase-9, caspase-3, and caspase-7 in a dose-dependent manner [131].

Corymbulosin A (**43**) (Figure 8) is a clerodane diterpene with an isozuelanin skeleton, isolated from the bark and the fruit of *Laetia corymbulosa* (Salicaceae). The compound showed significant cytotoxic activity against the human gliosarcoma cell line SF539, with an IC_50_ value of 0.6 μM after 48 h of treatment [132]. Subsequently, in another study corymbulosin A (**43**) was tested on five human cancer cell lines, where it confirmed its high cytotoxicity. The cell lines and the IC_50_ values were, respectively, IC_50_ of 0.45 μM (A549), IC_50_ of 0.43 μM (MDA-MB-231), IC_50_ of 0.44 μM (MCF-7), IC_50_ of 0.42 μM (KB), and IC_50_ of 0.45 μM (KB-VIN) [133].

Crassifolin U (**44**) (Figure 9) is a new clerodane diterpene from *Croton crassifolius* (Euphorbiaceae). The compound was tested for its cytotoxic activity on RAW 264.7 cells (macrophages) and found to be practically not cytotoxic at 50 μM after 24 h treatment. Thereafter, crassifolin U (**44**) was tested for its anti-inflammatory and anti-angiogenesis activities. Regarding the anti-inflammatory one, RAW 264.7 cells were pre-treated with crassifolin U (**44**) 50 μM for 12 h before stimulating the release of pro-inflammatory cytokines with lipopolysaccharides (LPS). After 12 h, the levels of IL-6 and tumour necrosis factor α (TNF-α) were evaluated with enzyme-linked immunoassay (ELISA) kits and crassifolin U (**44**) was able to decrease the levels of these pro-inflammatory cytokines, respectively, to 32.78% (IL-6) and 12.53% of those cells treated only with LPS. The anti-angiogenesis activity was assessed using tube formation assay in human umbilical vein endothelial cells (HUVECs) and crassifolin U (**44**) showed a strong anti-angiogenesis activity, reducing cavities formation, with IC_50_ values of 7.20 μM (junction densities), 48.27 μM (vessel percentage areas), and 8.62 μM (average vessel lengths) [134].

Clerodermic acid (**45**) (Figure 9) extracted from *Salvia nemorosa* (Lamiaceae) was tested on A549 (human lung epithelial cancer) cell line. The compound was able to reduce the cell viability in a concentration-dependent manner, with an IC_50_ value of 35 μg/mL at 48 h. Through the DNA ladder assay, it has been shown that this concentration produced DNA fragmentation and the DAPI staining demonstrated the presence of chromatin condensation and nuclear fragmentation, typical signs of apoptosis. Induction of apoptosis on treated A549 cells was confirmed by the Annexin V/Propidium iodide (PI) assay, which also showed the presence cells dead through necrosis. Moreover, the expression of HIF-1α and HIF-1β on treated cells was measured to evaluate the effect on cells adaptation to hypoxia. The treatment had no significant effects on the expression level of HIF-1β, but the compound was shown to inhibit cells adaptation to hypoxia through the downregulation of HIF-1α mRNA [135].

16-Hydroxycleroda-3,13-dien-15,16-olide (**46**) (Figure 9) isolated from *Polyalthia longifolia* (Annonaceae) was found to induce mitochondrial-dependent apoptosis in clear cell renal cell carcinoma (ccRCC) cells. The compound inhibited human kidney carcinoma (786-O and A-498) cells proliferation and arrested the cell cycle at the G2/M phase. After a 24 h treatment, the compound was able to downregulate cyclin-dependent kinase 1 (CDK1), CDK2, CDK4, cyclin B1, cyclin D1, and cyclin E1 in a concentration-dependent manner, to upregulate the tumour suppressor protein p53, to eliminate the CDK inhibitor p21 and to suppress the phosphorylation level of NF-κB and the expression of MMP-2, MMP-9, and VEGF [136]. Another study showed that compound 46 induced apoptosis on T24 human BC cells and caused loss of MMPs, increased ROS production and expression levels of cleaved caspase-3, cleaved PARP-1, phospho-Histone H2A.X (pH2A.X), arrested the cell cycle at G_1_ phase and reduced levels of CDK2, CDK4, and cyclin D1; conversely, it raised protein levels of p21 and p27Kip1. The authors also analysed the changes in gene expression caused by compound **46**, showing that 308 genes were up-regulated and 177 down-regulated. Of those up-regulated, 77 were involved in apoptosis and 23 in autophagy. On the other hand, the down-regulated ones were mainly involved in cell cycle regulation [137].

Compound **46** has also been found in *Justicia insularis* (Acanthaceae) in the form of a 1:1 mixture of 16-hydroxy epimers (α and β). Racemic mixture of compound **46** showed cytotoxic activity against ovarian cancer cell lines (OVCAR-4 and OVCAR-8). The inhibition of cells proliferation was calculated using sulforhodamine B (SRB) assay and the IC_50_ values were, respectively, 5.7 μM (OVCAR-4) and 4.4 μM (OVCAR-8). The compound was also tested on normal human ovarian surface epithelial (HOE) cells, where it was less cytotoxic (IC_50_ of 12.1 μM). The reduction in viability in treated cells is due to the induction of apoptosis, confirmed by the increased activities of caspase-3, capsase-7, caspase-8, and caspase-9 [138].

Crispene E (**47**) (Figure 9), from *Tinospora crispa* (Menispermaceae)*,* proved capable of inhibiting STAT3 dimerisation. This was demonstrated by evaluating its ability to disrupt the STAT3 binding to phosphorylated high-affinity peptide pYLPQTV-NH, with an IC_50_ of 10.27 μM. Its activity was also compared with that of pYLKTKF, reaching 80% of its activity, and with that of STAT3 inhibitor STA-21, reaching 210% of its activity at 100 μM. Compound **47** showed cytotoxic activity selectively against STAT3-dependent MDA-MB-231 breast cancer cell line (IC_50_ of 5.35 μM), whereas it was practically inactive towards STAT3 null A4 cancer cell line (IC_50_ > 100 μM against), suggesting a STAT3-specific inhibition. As confirmation of dimerisation inhibition activity, it was observed that compound **47** was able to down-regulate the mRNA expression of the STAT3 target genes bcl-2, cyclin D1, nicotinamide N-methyltransferase (NNMT), and fascin. The downregulation was particularly significant for bcl-2, which could be the main cause of the reduction in cell viability in MDA-MB-231. Furthermore, the specificity of the compound towards STAT3 was confirmed, noting that STAT1, a factor with tumour-suppressive properties, was not affected by its activity [139].

A new clerodane furano-diterpenoid, indicated as TC-2 (**48**) (Figure 9), and isolated in *Tinospora cordifolia* (Menispermaceae) exhibited anticancer activity against A549 (lung), PC-3 (prostate), SF-295 (central nervous system), MDA-MB-435 (Melanoma), HCT-116 (Colon), and MCF-7 (Breast) cancer cell lines. IC_50_ values were 8 μM (HCT-116) in colon carcinoma, 10.4 μM (PC-3) Prostate, 14.8 μM (MDA-MB-435) for melanoma, 23 μM (SF-295) in CNS carcinoma, 33 μM (A549) in lung cancer, and 40 μM (MCF-7) in breast carcinoma, highlighting how the HCT-116 cell line was the most susceptible to treatment. Further studies on the HCT-116 cell line have shown that compound **48** is capable of inducing apoptosis and chromatin condensation, fragmentation of nuclei, and significant externalisation of phosphatidylserine (PS). Mitochondrial membrane potential (MMP) changes were evaluated, highlighting a significant loss of MMP in HCT-116. In addition, a loss of mitochondrial function and release of cytochrome c to the cytosol were observed. These results suggest the induction of mitochondria-mediated apoptosis with activation of the initiator caspase 9, triggering the caspase cascade that results in DNA condensation/fragmentation and cell death. Furthermore, compound **48** causes an increase in ROS, which is related to mitochondrial damage, and induces autophagy in HCT-116 [140].

An epoxy clerodane diterpene, (5R,10R)-4R, 8R-dihydroxy-2S, 3R:15, 16-diepoxycleroda-13 (16), 17, 12S:18, 1S-dilactone (**49**) (Figure 9), named 8-Hydroxytinosporide, was isolated from *Tinospora cordifolia* (Menispermaceae) showed a potential anti-proliferative effect in MCF-7 cells (IC_50_ 3.2 at 24 h and 2.4 μM at 48 h) by down-regulating Mdm2 and stimulating the expression of tumour suppressor genes cyclin-dependent kinase Inhibitor 2A (Cdkn2A), retinoblastoma protein 1 (pRb1), and p53, which increased expression, activates the pro-apoptotic protein Bax. Furthermore, the compound significantly increases the level of ROS in the treated cells in a dose-dependent manner. Moreover, the epoxy clerodane diterpene activates cyclin-dependent kinase Inhibitor 1A (Cdkn1A) and inhibits proliferating cell nuclear antigen (PCNA) [141].

Teotihuacanin (**50**) (Figure 9) is a clerodane present in *Salvia amarissima* (Lamiaceae), characterised by an unusual structure with a new carbon skeleton containing a spiro-10/6 bicyclic system. Its cytotoxic activity on human breast (MCF-7 and MDA-MB-231), colon (HCT-15 and HCT-116), and cervix (HeLa) cancer cell lines was tested. The compound showed moderate cytotoxicity against MDA (IC_50_ of 12.3 μg/mL), HeLa (IC_50_ of 13.7 μg/mL), HCT-15 (IC_50_ of 12.9 μg/mL), and HCT-116 (IC_50_ of 10.9 μg/mL), but no cytotoxicity was demonstrated against MCF-7 (IC_50_ > 20 μg/mL). The most interesting feature of this molecule is its powerful activity as modulator of multidrug resistance (MDR) in MCF-7 cells, suggesting that teotihuacanin (**50**) can be used as a starting compound for the synthesis of more potent MDR modulators for cancer chemo-therapy [142]. A summary of the main anti-cancer activities of clerodane diterpenes is shown in Table 3.

## 7. Conclusions

The pharmacological treatment of cancer has made considerable signs of progress thanks to modern anti-tumour drug research. In fact, nowadays some tumours can be successfully cured and, in several others, long-term patient survival has been achieved.

On the flip side, the growing incidence of cancer with the limitations of conventional therapies mainly including the high cost and high toxicity of the actually used anti-cancer drugs represents an important challenge to researchers to develop an alternative, effective, less toxic, eco-sustainable, and cost-effective strategy for the treatment of this high-social-impact disease. In this context, natural compounds have made a substantial contribution to cancer therapy and anticancer drug discovery, considering that many of these are now first-choice treatments in clinical oncology practice. This review article provides information on natural diterpenes with an abietane, clerodane, and labdane skeleton as potential new agents for the treatment of different cancer types. The studies examined above suggest that natural abietane, clerodane, and labdane diterpenes possess anti-cancer potential through different mechanisms of action such as anti-proliferative activity, induction of differentiation, anti-angiogenesis, pro-apoptosis, and inhibition of adhesion, migration, invasion, and metastasis by aiming at cellular targets specifically expressed in cancer cells. Moreover, the analysis of the studies published in recent years shows also a great interest in the evaluation of a potential synergism of action among the most known natural anti-cancer diterpenes and known anti-tumour drugs (e.g., doxorubicin, topotecan, and 5-fluorouracil), though clinical data demonstrating relevant advantages are still lacking.

Concluding, these natural diterpenes represented, during the last years, an active field of research producing a high number of works on isolation, structural characterisation, and analysis of their bioactivity, and may be considered one of the promising leading classes of therapeutic molecules in cancer treatment. However, it is important to highlight that further preclinical and clinical studies are needed before their reliable use as anticancer and chemopreventive drugs.

## Figures and Tables

**Figure 1 molecules-27-04791-f001:**
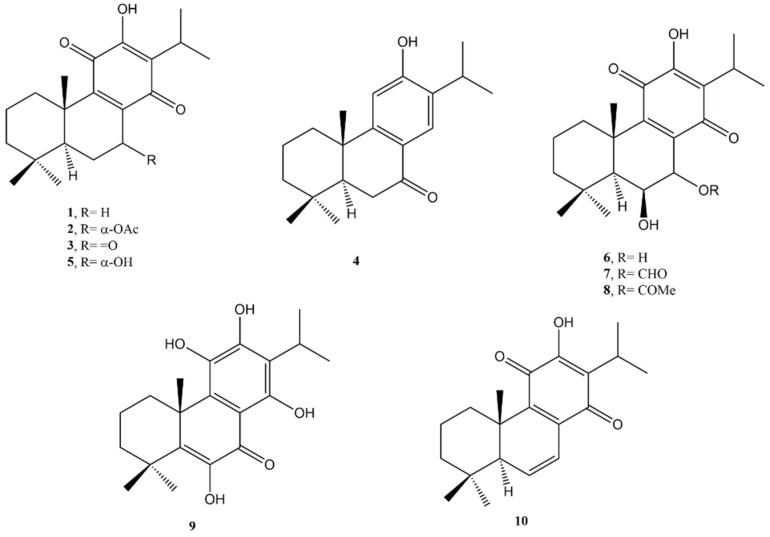
Bioactive abietane diterpenes: Royleanone (**1**); 7*α*-acetoxyroyleanone (**2**); 7-ketoroyleanone (**3**); Sugiol (**4**); Horminone (**5**); 7*α*,6*β*-dihydroxyroyleanone (**6**); 7*α*-formyloxy-6*β*-hydroxyroyleanone (**7**); 7*α*-acetoxy-6*β*-hydroxyroyleanone (**8**); Coleon U (**9**); 6,7-dehydroroyleanone (**10**).

**Figure 2 molecules-27-04791-f002:**
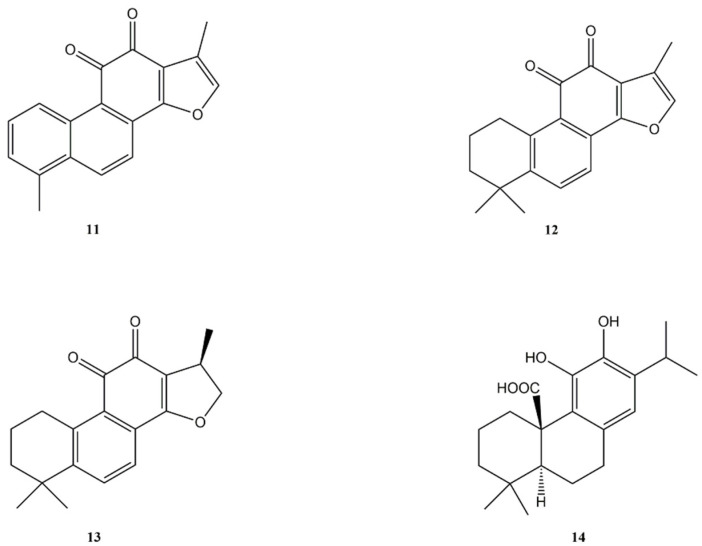
Bioactive abietane diterpenes: Tanshinone I (**11**); Tanshinone IIA (**12**); Cryptotanshinone (**13**); Carnosic acid (**14**).

**Figure 3 molecules-27-04791-f003:**
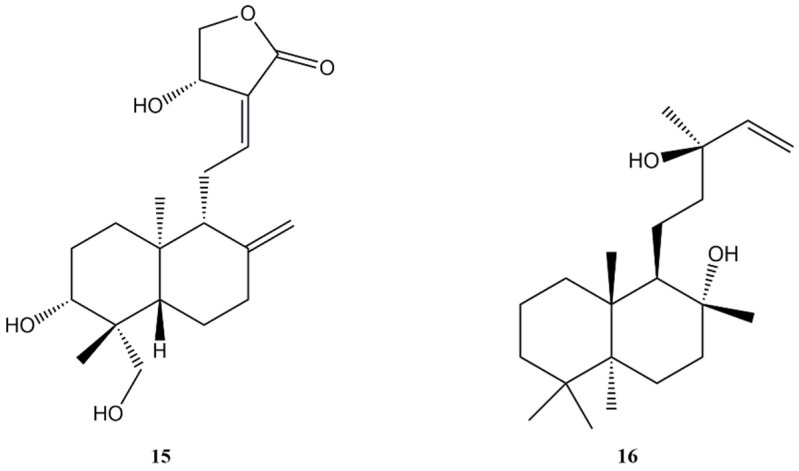
Bioactive labdane diterpenes: Andrographolide (**15**); Sclareol (**16**).

**Figure 4 molecules-27-04791-f004:**
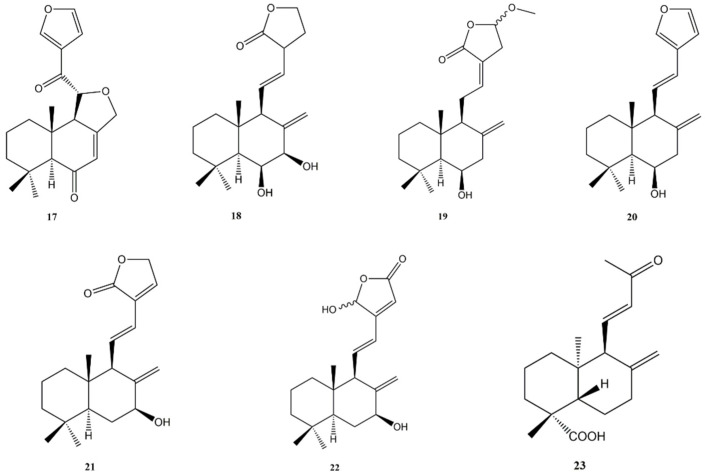
Bioactive labdane diterpenes: Hedylongnoid A (**17**); Hedylongnoid B (**18**); Hedylongnoid C (**19**); Yunnancoronarin A (**20**); Hedyforrestin B (**21**); Hedyforrestin C (**22**); (1R,4aS,5R,8aS)-1,4a-dimethyl-5-[(1E)-3-oxobut-1-en-1-yl]decahydronaphthalene-1-carboxylic acid (**23**).

**Figure 5 molecules-27-04791-f005:**
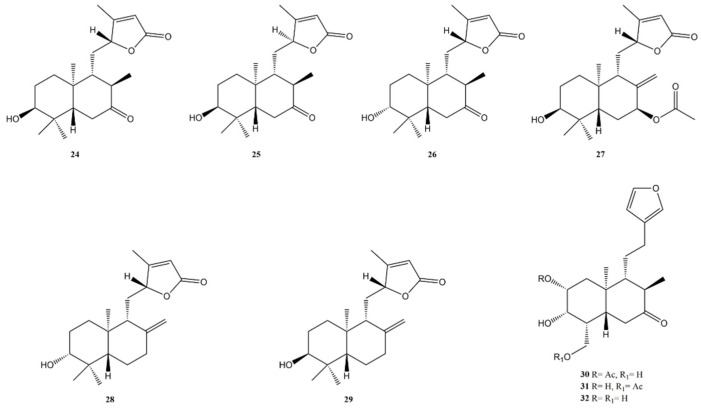
Bioactive ent-labdane diterpenes: Uasdlabdanes A-F (**24**–**29**); Grazielabdane B (**30**); Grazielabdane C (**31**); Grazielabdane A (**32**).

**Figure 6 molecules-27-04791-f006:**
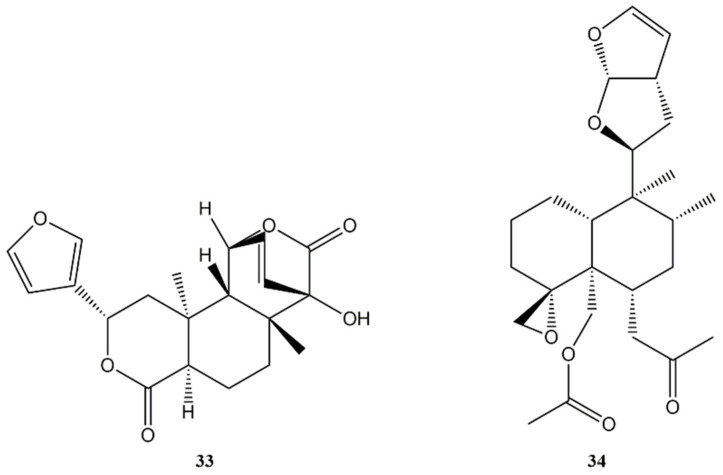
Structures of bioactive *trans* and *cis* clerodane diterpenes: Columbin (**33**); Clerodin (**34**).

**Figure 7 molecules-27-04791-f007:**
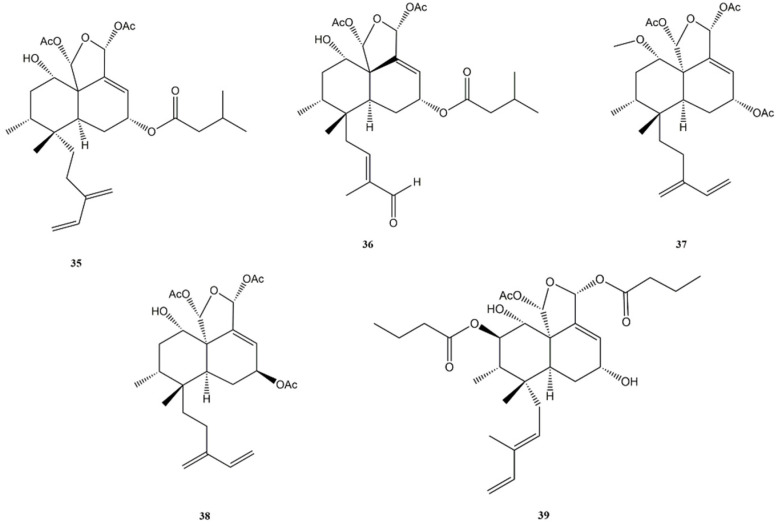
Bioactive clerodane diterpenes: Corymbulosin X (**35**); Caseakurzin B (**36**); Corymbulosin M (**37**); Kurzipene D (**38**); Casearin D (**39**).

**Figure 8 molecules-27-04791-f008:**
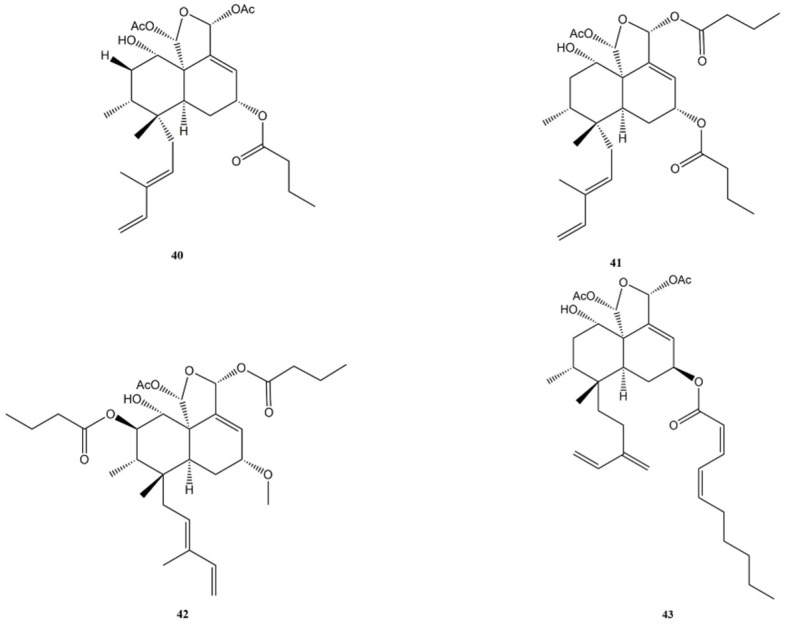
Bioactive clerodane diterpenes: Caseargrewiin F (**40**); Casearin X (**41**); Casearin J (**42**); Corymbulosin A (**43**).

**Figure 9 molecules-27-04791-f009:**
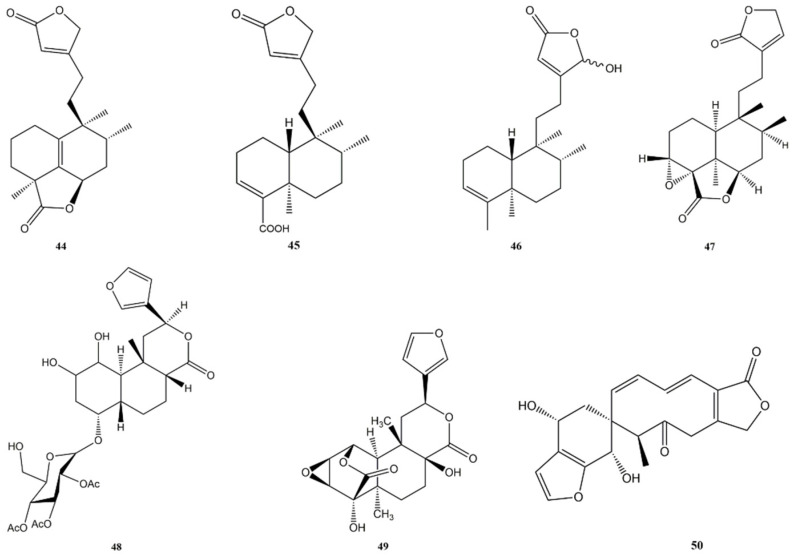
Bioactive clerodane diterpenes: Crassifolin U (**44**); Clerodermic acid (**45**); 16-Hydroxycleroda-3,13-dien-15,16-olide (**46**); Crispene E (**47**); TC-2 (**48**); 8-Hydroxytinosporide (**49**); Teotihuacanin (**50**).

**Table 1 molecules-27-04791-t001:** Sources and biological activities on cancer cell lines of abietane diterpenes.

N°	Name	Source	Biological Activity	Refs.
**1**	Royleanone	*Peltodon longines* (Lamiaceae)	MIA PaCa2, LNCaP cells (IC_50_ = 12.5 μM) Alkylating properties; human topoisomerase I inhibition; apoptosis and cell cycle arrest; inhibition of Akt PI3K-mTOR pathway	[19,22]
**2**	7α-acetoxyroyleanone	*Peltodon longines* (Lamiaceae)	MIA PaCa2 cells (IC_50_ = 4.7 μM) alkylating properties; inhibitors of the human topoisomerase I	[19]
**3**	7-ketoroyleanone	*Peltodon longines* (Lamiaceae)	MIA PaCa2 cells alkylating properties; inhibitors of the human topoisomerase I–II	[19]
**4**	Sugiol	*Peltodon longines* (Lamiaceae) *Taxodium distichum* (Cupressaceae)	MIA PaCa2, PANC-1 cells (IC_50_ = 9.0 μM) alkylating properties; inhibitors of the human topoisomerase I–II	[19,21]
**5**	Horminone	*Peltodon longines* (Lamiaceae)	MIA PaCa2 cells alkylating properties; inhibitors of the human topoisomerase I	[19]
**6**	Dihydroxyroyleanone	*Plectranthus madagascariensis* (Lamiaceae)	HCT116, MDA-MB-231, MCF-7, NCI-H460, NCI-H460/R cells anti-proliferative effect	[23]
**7**	7α-formyloxy-6β-hydroxyroyleanone	*Plectranthus madagascariensis* (Lamiaceae)	HCT116, MDA-MB-231, MCF-7, NCI-H460, NCI-H460/R cells anti-proliferative effect	[23]
**8**	7α-acetoxy-6β-hydroxyroyleanone	*Plectranthus madagascariensis* (Lamiaceae)	HCT116, MDA-MB-231, MCF-7 (GI_50_ = 6.4 μM), NCI-H460 (GI_50_ = 2.7 μM), NCI-H460/R (GI_50_ = 3.1 μM) cells anti-proliferative effect	[23]
**9**	Coleon U	*Plectranthus madagascariensis* (Lamiaceae)	HCT116, MDA-MB-231, MCF-7 (GI_50_ =5.5 μM), NCI-H460, NCI-H460/R cells anti-proliferative effect	[23]
**10**	6,7-dehydroroyleanone	*Plectranthus madagascariensis* *Plectranthus**ecklonii* (Lamiaceae)	Glioma cells apoptosis; cell cycle arrest	[24,25]
**11**	Tanshinone I	*Salvia miltiorrhiza* (Lamiaceae)	HEC-1-A cells (IC_50_ = 20 μM) apoptosis; ROS levels increase; JAK/STAT pathway modulation	[26]
**12**	Tanshinone IIA	*Salvia miltiorrhiza* (Lamiaceae)	HCT116, MDA-MB-231, MCF-7, NCI-H460, NCI-H460/R, BxPC-3, PC9, H1299, HeLa, SGC7901 cells apoptosis; ER stress; cycle cellular modulation; reversing the multidrug resistance; signalling pathway and protein expression regulation	[27,33,35,50]
**13**	Cryptotanshinone	*Salvia miltiorrhiza* (Lamiaceae)	HCT116, MDA-MB-231, MCF-7, HCCC-9810, DU-145, Rh30, K562/ADM, HeLa, HT29, SW620, SW480 cells apoptosis, cycle cellular arrest; reversing the multidrug resistance; signalling pathway and protein expression regulation	[51,66]
**14**	Carnosic acid	*Salvia* *officinalis* *Rosmarinus officinalis* (Lamiaceae)	Glioma cells, MDA-MB-231; ERBB2^+^; AGS, MKN-45, KYSE-150, HepG2, SMMC-7721, DU145, PC3, A-549, NCI-H460 cells apoptosis, cycle cellular arrest; ROS level increase; autophagy stimulation; signalling pathway regulation, p62 down regulation; p-AKT inhibition; reversing the multidrug resistance; NRF2 stabilisation;	[78,91]

Abbreviation: A549, human lung cancer cell line; AGS, human caucasian gastric adenocarcinoma; Akt, protein kinase B; BxPC-3, human pancreatic cancer cell; C9, human lung cancer cell line; DU-145, human prostate carcinoma cell line; ER, endoplasmic reticulum;ERBB2+, human breast cancer line; GC-7901, human gastric cancer cell line; H1299, human lung cancer cell line; HCCC-9810, human cholangiocarcinoma cell line; HCT-116, colon carcinoma cell line; HEC-1-A, endometrial carcinoma; HeLa, cervical cancer cell line; S; HepG-2, human liver cancer cell line; HT29, human colorectal cancer cell line; JAK, janus kinase; K562/ADM, human chronic myeloid leukemia cell line; KYSE-150, esophageal squamous cell carcinoma; ROS, reactive oxygen species; LNCaP, human prostate carcinoma cell line; MCF-7, human breast cell line; MDA-MB-231, human breast adenocarcinoma cell line; MIA PaCa2, human pancreatic cancer cell line; MKN-45, human gastric cancer cell line; mTOR, mammalian target of rapamycin; NCI-H460, human lung cancer cell line; NRF2, nuclear factor erythroid 2-related factor 2; p62, ubiquitin-binding protein p62; p-AKT, phosphor protein kinase B; PANC-1, human pancreas epithelioid carcinoma cell line; PC3, caucasian prostate adenocarcinoma cell line; PI3K, phosphatidylinositol 3-kinase; Rh30, rhabdomyosarcoma cell line; SMMC-7721, human liver cancer cell line; STAT, signal transducer and activator of transcription protein; SW480 cells, human colon cancer cell line; SW620, human colon cancer cell line.

**Table 2 molecules-27-04791-t002:** Sources and biological activities on cancer cell lines of labdane diterpenes.

N°	Name	Source		Refs.
**15**	Andrographolide	*Andrographis paniculata* (Acanthaceae) *Swertia pseudochinensis* (Gentianaceae)	HCT-116, U937, SGC7901 (IC_50_= 38 μM), AGS (IC_50_ = 44 μM), HeLa, MCF-7 cells apoptosis; cell cycle arrest; ROS levels increase; pathway modulation	[99,107]
**16**	Sclareol	*Salvia sclarea* (Lamiaceae)	HCT-116, HeLa, MCF-7, MDA-MB-469, CaCo-2, 4T1, H1688 cells apoptosis; cell cycle arrest; ROS levels; DNA damage increase	[108,109]
**17**	Hedylongnoids A	*Hedychium longipetalum* (Zingiberaceae)	HeLa, SGC-7901 cells: no cytotoxic	[120]
**18**	Hedylongnoids B	*Hedychium longipetalum* (Zingiberaceae)	HeLa, SGC-7901 cells: no cytotoxic	[120]
**19**	Hedylongnoids C	*Hedychium longipetalum* (Zingiberaceae)	HeLa, SGC-7901 (IC_50_ = 8.74 μg/mL) cells anti-proliferative effect	[120]
**20**	yunnancoronarin A	*Hedychium longipetalum* (Zingiberaceae)	HeLa (IC_50_ = 6.58 μg/mL), SGC-7901 (IC_50_ = 6.21 μg/mL) cells anti-proliferative effect	[120]
**21**	hedyforrestin B	*Hedychium longipetalum* (Zingiberaceae)	HeLa, SGC-7901 cells anti-proliferative effect	[120]
**22**	hedyforrestin C	*Hedychium longipetalum* (Zingiberaceae)	HeLa, SGC-7901 (IC_50_ = 7.29 μg/mL) cells anti-proliferative effect	[120]
**23**	(1R,4aS,5R,8aS)-1,4a-dimethyl-5-[(1E)-3-oxobut-1-en-1-yl]decahydronaphthalene-1-carboxylic acid	*Juniperus oblonga* (Cupressaceae)	MCF-7, HeLa, HepG2 cells weak cytotoxic	[121]
**24**	Uasdlabdanes A	*Eupatorium obtusissmum* (Asteraceae)	HBL-100, T-47D, HeLa, A549, SW1573, WiDr cells anti-proliferative effect	[122]
**25**	Uasdlabdanes B	*Eupatorium obtusissmum* (Asteraceae)	HBL-100, T-47D, HeLa, A549, SW1573, WiDr cells anti-proliferative effect	[122]
**26**	Uasdlabdanes C	*Eupatorium obtusissmum* (Asteraceae)	HBL-100, T-47D, HeLa, A549, SW1573, WiDr cells anti-proliferative effect	[122]
**27**	Uasdlabdanes D	*Eupatorium obtusissmum* (Asteraceae)	HBL-100, T-47D, HeLa, A549, SW1573, WiDr cells anti-proliferative effect	[122]
**28**	Uasdlabdanes E	*Eupatorium obtusissmum* (Asteraceae)	HBL-100, T-47D, HeLa, A549, SW1573, WiDr cells anti-proliferative effect	[122]
**29**	Uasdlabdanes F	*Eupatorium obtusissmum* (Asteraceae)	HBL-100, T-47D, HeLa, A549, SW1573, WiDr cells anti-proliferative effect	[122]
**30**	Grazielabdane B	*Grazielia gaudichaudeana* (Asteraceae)	OVCAR-03 (IC_50_ = 5.5 μM), U251 (IC_50_ = 25.63 μM) cells anti-proliferative effect	[95]
**31**	Grazielabdane C	*Grazielia gaudichaudeana* (Asteraceae)	OVCAR-03, U251 cells: antiproliferative	[95]
**32**	Grazielabdane A	*Grazielia gaudichaudeana* (Asteraceae)	OVCAR-03 (IC_50_ = 5.5 μM), U251 (IC_50_ = 25.63 μM) cells anti-proliferative	[95]

Abbreviation: 4T1, breast cancer cell line; A549, human lung cancer cell line; AGS, human caucasian gastric adenocarcinoma; CaCo-2, colorectal adenocarcinoma cell line; H1688, human lung cancer cell line; HBL-100, human breast cancer cell line; HCT-116, Colon carcinoma cell line; HeLa, cervical cancer cell line; HepG-2, human liver cancer cell line; IC_50_, half maximal inhibitory concentration; MCF-7, human breast cell line; MDA-MB-469, human metastatic breast adenocarcinoma cell line; OVCAR-03, ovary cell line; ROS, reactive oxygen species; SGC-7901, human gastric cancer cell line; SW1573, human lung cancer cell line; T-47D, human breast cancer cell line; U251, glioma cell line; U937, human macrophage cell line; WiDr, human colon cancer cell line.

**Table 3 molecules-27-04791-t003:** Sources and biological activities on cancer cell lines of clerodane diterpenes.

N°	Name	Source	Biological Activity	Refs.
**35**	Corymbulosin X	*Anacolosa clarkii* (Olacaceae)	A-673 (TGI_50_ = 0.7 μM), SJCRH30 (TGI_50_ =0.34 μM), D283 (TGI_50_ = 0.36 μM), Hep293TT (TGI_50_ = 0.22 μM) cells cytotoxic effect	[124]
**36**	Caseakurzin B	*Casearia kurzii* (Salicaceae)	A549 cells (IC_50_ = 4.4 μM) apoptosis; cell cycle arrest	[125]
**37**	Corymbulosin M	*Casearia kurzii* (Salicaceae)	A549 (IC_50_ = 5.5 μM), HeLa (IC_50_ = 4.1 μM), HepG2 (IC_50_ = 9.3 μM) cells apoptosis; cell cycle arrest	[126]
**38**	Kurzipene D	*Casearia kurzii* (Salicaceae)	A549 (IC_50_ = 10.9 μM), HeLa (IC_50_ = 12.4 μM), HepG2 (IC_50_ = 9.7 μM), K562 (IC_50_ = 7.2 μM) cells apoptosis; cell cycle arrest	[127]
**39**	Casearin D	*Casearia sylvestris* (Salicaceae)	HL-60 (IC_50_ = 3.44 μM), MDA-MB-231 (IC_50_ = 4.23 μM), Hs578-T (IC_50_ = 4.39 μM), MX-1 (IC_50_ = 6.50 μM), PC-3 (IC_50_ = 1.41 μM), DU145 (IC_50_ = 8.53 μM), B-16/F10 (IC_50_ = 6.52 μM) cells anti-proliferative effect; cell cycle arrest	[127]
**40**	Caseargrewiin F	*Casearia sylvestris* (Salicaceae)	HL-60 (IC_50_ = 0.20 μM), MDA-MB-231 (IC_50_ = 0.14 μM), Hs578-T (IC_50_ = 0.26 μM), MX-1 (IC_50_ = 0.36 μM), PC-3 (IC_50_ = 0.31 μM), DU145 (IC_50_ = 0.76 μM), B-16/F10 (IC_50_ = 0.16 μM) cells anti-proliferative effect; apoptosis; cell cycle arrest; DNA fragmentation	[128,130]
**41**	Casearin X	*Casearia sylvestris* (Salicaceae)	HL-60 (IC_50_ = 0.28 μM), MDA-MB-231 (IC_50_ = 1.51 μM), Hs578-T (IC_50_ = 1.14 μM), MX-1 (IC_50_ = 0.95 μM), PC-3 (IC_50_ = 0.86 μM), DU145 (IC_50_ = 1.19 μM), B-16/F10 (IC_50_ = 1.15 μM) cells anti-proliferative effect; apoptosis; cell cycle arrest; DNA fragmentation	[128,130]
**42**	Casearin J	*Casearia sylvestris* (Salicaceae)	T-ALL, CCRF-CEM (IC_50_ = 0.7 μM), CEM-ADR500 cells apoptosis; oxidative stress; DNA fragmentation	[131]
**43**	Corymbulosin A	*Laetia corymbulosa* (Salicaceae)	SF539 (IC_50_ = 0.6 μM), A549 (IC_50_ = 0.45 μM), MDA-MB-231 (IC_50_ = 0.43 μM), MCF-7 (IC_50_ = 0.44 μM), KB (IC_50_ = 0.42 μM), KB-VIN (IC_50_ = 0.45 μM) cells anti-proliferative effect	[132,133]
**44**	Crassifolin U	*Croton crassifolius* (Euphorbiaceae)	RAW 264.7, HUVECs cells anti-inflammatory, anti-angiogenesis	[134]
**45**	Clerodermic acid	*Salvia nemorosa* (Lamiaceae)	A549 (IC_50_ = 35 μg/mL) cells Apoptosis	[135]
**46**	16-Hydroxycleroda-3,13-dien-15,16-olide	*Polyalthia longifolia* (Annonaceae) *Justicia insularis* (Acanthaceae)	ccRCC, 786-O, A498, OVCAR-4 (IC_50_ = 5.7 μM), OVCAR-8 (IC_50_ = 4.4 μM) cells apoptosis; cell cycle arrest; down regulation cycline dependent; pathway and protein expression regulation	[136,138]
**47**	Crispene E	*Tinospora crispa* (Menispermaceae)	MDA-MB-231 (IC_50_ = 5.35 μM) cells anti-proliferative effect; apoptosis; pathway regulation (STAT3)	[139]
**48**	TC-2	*Tinospora cordifolia* (Menispermaceae)	A549 (IC_50_ = 33 μM), PC-3 (IC_50_ = 10 μM), SF295 (IC_50_ = 4.4 μM), MDA-MB-435 (IC_50_ = 14.8 μM), MCF-7 (IC_50_ = 4.4 μM), HCT-116 (IC_50_ = 8 μM) cells anti-proliferative effect; apoptosis; ROS levels increase	[140]
**49**	8-Hydroxytinosporide	*Tinospora cordifolia* (Menispermaceae)	MCF-7 (IC_50_ = 2.4 μM) cells antiproliferative effect; ROS levels increase; expression of tumour suppressor genes	[141]
**50**	Teotihuacanin	*Salvia amarissima* (Lamiaceae)	MDA-MB-231 (IC_50_ = 12.3 μg/mL), HCT-15 (IC_50_ = 12.9 μg/mL), HCT116 (IC_50_ = 10.9 μg/mL), HeLa (IC_50_ = 13.7 μg/mL) cells anti-proliferative effect; modulator of multidrug resistance	[142]

Abbreviation: 786-O, human kidney adenocarcinoma cell line; A-498, human kidney carcinoma cell line; A-549 cells, human lung cancer cell line; A-673, Ewing’s sarcoma cell line; B16-F10, murine melanoma cell line; ccRCC cells, clear cell renal cell carcinoma cell line; CCRF-CEM, human acute lymphoblastic leukemia T-lymphoblasts cell line; CEM-ADR5000, human acute lymphoblastic leukemia drug-resistant T-lymphoblasts cell line; D283, medulloblastoma cell line; DU-145, human prostate carcinoma cell line; HCT-116, colon carcinoma cell line; HCT-15, human colon adenocarcinoma cell line; HeLa, cervical cancer cell line; Hep293TT, hepatoblastoma cell line; HepG-2, human liver cancer cell line; HL-60, caucasian promyelocytic leukemia cell line; Hs578-T, human breast carcinoma cell line; HUVEC, human umbilical vein endothelial cell; IC50, half maximal inhibitory concentration; K562, human chronic myeloid leukemia cell line; KB-VIN, oral carcinoma vincristine-resistant cell line; MCF-7, human breast cell line; MDA-MB-231, human breast adenocarcinoma cell line; MX-1, human breast carcinoma cell line; OVCAR-4, human ovarian adenocarcinoma cell line; OVCAR-8, human ovarian adenocarcinoma cell line; PC3, caucasian prostate adenocarcinoma cell line; RAW 264.7, Abelson murine leukemia monocyte/macrophage cell line; ROS, reactive oxygen species; SF295, human glioblastoma cell line; SF539, human gliosarcoma cell line; SJCRH30, rhabdomyosarcoma cell line; T-ALL, T-cell acute lymphoblastic leukemia; TGI, total growth inhibition.

## Data Availability

Not applicable.

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
