# Peer review of "Advances on Natural Abietane, Labdane and Clerodane Diterpenes as Anti-Cancer Agents: Sources and Mechanisms of Action"

_molecules, 2022, doi:10.3390/molecules27154791_

Round 1

Reviewer 1 Report

The manuscript entitled “Advances on Natural Abietane, Labdane and Clerodane Diterpenes as Anti-Cancer Agents: Sources and Mechanisms of Action” mainly reviewed a promising class of phytochemicals such as diterpenes with abietane, clerodane, and labdane skeleton and these natural source, mechanisms. The manuscript reviews articles from January 2017 to March 2022. I admit the value of this article but there are still some issues to be addressed.

1. This manuscript should add some tables to summarize the structure, nomenclature, bioactivity and corresponding references of each class of diterpenes, so that readers can find relevant contents quickly.

2. This manuscript relates some diterpenes not has clearly plant sources, should make it complete or illustrate with an example.

3. The manuscript should be emphasized describe the relation of these diterpenes. If the commonality of the three diterpenes is not be described, the article will lack logic.

4. The structure of some compounds needs to be modified, such as “7-ketoroyleanone (3)”, the structure of “R” has an ambiguity.

Author Response

REVIEWER 1

The manuscript entitled “Advances on Natural Abietane, Labdane and Clerodane Diterpenes as Anti-Cancer Agents: Sources and Mechanisms of Action” mainly reviewed a promising class of phytochemicals such as diterpenes with abietane, clerodane, and labdane skeleton and these natural source, mechanisms. The manuscript reviews articles from January 2017 to March 2022. I admit the value of this article but there are still some issues to be addressed.

A: We thank the reviewer for his/her kind comments and suggestions to improve the quality of our work. We have studied your comment carefully and have made the expected corrections, which we hope will meet your approval. Revised portions are highlighted in red in the manuscript.

1 This manuscript should add some tables to summarize the structure, nomenclature, bioactivity and corresponding references of each class of diterpenes, so that readers can find relevant contents quickly.

A: The authors accepted the reviewer's suggestion and added the tables

2 This manuscript relates some diterpenes not has clearly plant sources, should make it complete or illustrate with an example.

A: The authors added plant sources and examples in the manuscript (page 14 line 608: “Compound 33 has been isolated in several plant such as Spenocentrum jollyanum Pierre (Menispermaceae) and Jateorhiza columba Miers (Menispermaceae) while the compound 34 was found in Clerodendrum infortunatum L. (Lamiaceae)”

3 The manuscript should be emphasized describe the relation of these diterpenes. If the commonality of the three diterpenes is not be described, the article will lack logic.

A: The authors met the reviewer's suggestion and added the sentence in page 2 line 65: “The purpose of this review article is to highlight the potential anti-cancer activities and sources of a natural class of diterpenes with abietane, labdane, and clerodane skeletons belonging to the group of furanoditerpenoids. This particular group of rare diterpenoids presents a furan ring, which confers a significant biological activity due to the aromatic system that allows the formation of hydrogen bonds and hydrophobic interactions with cellular components.

4 The structure of some compounds needs to be modified, such as “7-ketoroyleanone (3)”, the structure of “R” has an ambiguity.

A: According to the reviewer's suggestion compounds structure were revised

Reviewer 2 Report

The article is interesting but it needs several chemical improvements:

In structures 1-8, 35 there needs to be added the hydrogen in pos. C-5 to show the connection of the rings, as in (10).

Cryptotanshinone needs to have shown the stereochemistry of methyl on the five-membered ring.

Names of Hedylongnoids A (17); Hedylongnoids B (18); Hedy-longnoids C (19); in Fig. 4 shall be in singular.

I would name the compound (23) as (1R,4aS,5R,8aS)-1,4a-dimethyl-5-[(1E)-3-oxobut-1-en-1-yl]decahydronaphthalene-1-carboxylic acid

I would suggest checking the structure of Columbin CASRN 546-97-4. It is not possible to connect two stereogenic centres by a wedge bond. Also compound (50) Teotihuacanin shall be checked with Teotihuacanin CASRN 1796568-88-1

I was not able to find in Chemical Abstracts the structures (37), (44), (46), and (38)

In structure 41 the hydrogen on C-2 has no meaning.

Chemical Abstracts show Corymbulosin A 326473-51-3 with both ester double bonds as cis

For compound (49) Chemical Abstracts give name 8-Hydroxytinosporide

As for the compound TC-2, In Chemical Abstracts I found only 1-Deacetyltinosineside A , CAS RN 1253639-70-1.

Author Response

REVIEWER 2

The article is interesting but it needs several chemical improvements:

A: We thank the reviewer for his/her kind comments and suggestions to improve the quality of our work. We have studied your comment carefully and have made the expected corrections, which we hope will meet your approval. Revised portions are highlighted in red in the manuscript.

1 In structures 1-8, 35 there needs to be added the hydrogen in pos. C-5 to show the connection of the rings, as in (10).

A: According to the reviewer's suggestion compound structure was revised

2 Cryptotanshinone needs to have shown the stereochemistry of methyl on the five-membered ring.

A: According to the reviewer's suggestion compound structure was revised

3 Names of Hedylongnoids A (17); Hedylongnoids B (18); Hedy-longnoids C (19); in Fig. 4 shall be in singular.

A: The authors met the reviewer’s suggestion

4 I would name the compound (23) as (1R,4aS,5R,8aS)-1,4a-dimethyl-5-[(1E)-3-oxobut-1-en-1-yl]decahydronaphthalene-1-carboxylic acid

A: According to the reviewer's suggestion the author changed the name from (4R,5S,9S,10R)-13-des-ethyl-13-oxolabda-8(17),11E-dien-19-oic acid to (1R,4aS,5R,8aS)-1,4a-dimethyl-5-[(1E)-3-oxobut-1-en-1-yl]decahydronaphthalene-1-carboxylic acid

5 I would suggest checking the structure of Columbin CASRN 546-97-4. It is not possible to connect two stereogenic centres by a wedge bond. Also compound (50) Teotihuacanin shall be checked with Teotihuacanin CASRN 1796568-88-1

A: According to the reviewer's suggestion compounds structure were revised

6 I was not able to find in Chemical Abstracts the structures (37), (44), (46), and (38)

A: According to the reviewer's suggestion compound: 37 CASRN: 2219352-11-9; 38 CASRN: 2522596-01-4; 46 CASRN: 141979-19-3 were checked. For compound 44 the structure is from the relative reference (134)

7 In structure 41 the hydrogen on C-2 has no meaning.

A: According to the reviewer's suggestion compound structure was revised

8 Chemical Abstracts show Corymbulosin A 326473-51-3 with both ester double bonds as cis

A: According to the reviewer's suggestion compound structure was revised

9 For compound (49) Chemical Abstracts give name 8-Hydroxytinosporide

A: The Authors met the reviewer's suggestion

10 As for the compound TC-2, In Chemical Abstracts I found only 1-Deacetyltinosineside A , CAS RN 1253639-70-1.

A: For compound 48 (TC-2), the structure does not match the name  suggested by  the reviewer

Reviewer 3 Report

The presented manuscript reviews on the recent literature focused on anticancer activity studies of certain classes of diterpenes having abietane, clerodane, and labdane skeleton. The work summarizes all the most important publications in this field and is a rich source of information for chemists dealing with natural compounds with significant biological activity. I find it very valuable that the authors deliberately excluded from their review many publications dealing with the study of the biological activity of various plant extracts and other mixtures with an unspecified composition, taking into account only well-defined chemical entities with a proven structure.

The manuscript was written efficiently, in a clear scientific language, and the considerable amount of information it contains is a serious scientific asset. Accordingly, I recommend that the manuscript be published.

- minor points:

- page 5, Fig. 2, the absolute stereochemistry of the methyl group in compound 13 (Cryptotanshinone) should be shown (this is well-known compound)

- page 7, line 297, correct “strckly”

- page 14, line 605, the sentence „Based on the absolute stereochemistry to the relative configuration…” is not clear

- page 18, Fig. 9, the stereochemistry of compound 49 (Epoxy clerodane diterpene) should be presented

Author Response

REVIEWER 3

The presented manuscript reviews on the recent literature focused on anticancer activity studies of certain classes of diterpenes having abietane, clerodane, and labdane skeleton. The work summarizes all the most important publications in this field and is a rich source of information for chemists dealing with natural compounds with significant biological activity. I find it very valuable that the authors deliberately excluded from their review many publications dealing with the study of the biological activity of various plant extracts and other mixtures with an unspecified composition, taking into account only well-defined chemical entities with a proven structure. The manuscript was written efficiently, in a clear scientific language, and the considerable amount of information it contains is a serious scientific asset. Accordingly, I recommend that the manuscript be published.

- minor points:

A: We thank the reviewer for his/her kind comments and suggestions to improve the quality of our work. We have studied your comment carefully and have made the expected corrections, which we hope will meet your approval. Revised portions are highlighted in red in the manuscript.

1 page 5, Fig. 2, the absolute stereochemistry of the methyl group in compound 13 (Cryptotanshinone) should be shown (this is well-known compound)

A: The authors met the reviewer's suggestion and showed stereochemistry of the methyl group in compound 13

2 page 7, line 297, correct “strckly”

A: The authors corrected strckly with strictly

3 page 14, line 605, the sentence „Based on the absolute stereochemistry to the relative configuration…” is not clear

A: The authors met the reviewer's suggestion and modified the sentence

4 page 18, Fig. 9, the stereochemistry of compound 49 (Epoxy clerodane diterpene) should be presented

A: The authors added the stereochemistry of compound 49 (Epoxy clerodane diterpene)